# Is Smoothness the Key to Robustness? A Comparison of Attention and Convolution Models Using a Novel Metric

## Abstract

Robustness is a critical aspect of machine learning models. Existing robustness evaluation approaches often lack theoretical generality or rely heavily on empirical assessments, limiting insights into the structural factors contributing to robustness. Moreover, theoretical robustness analysis is not applicable for direct comparisons between models. To address these challenges, we propose *TopoLip*, a metric based on layer-wise analysis that bridges topological data analysis and Lipschitz continuity for robustness evaluation. TopoLip provides a unified framework for both theoretical and empirical robustness comparisons across different architectures or configurations, and it reveals how model parameters influence the robustness of models. Using TopoLip, we demonstrate that attention-based models typically exhibit smoother transformations and greater robustness compared to convolution-based models, as validated through theoretical analysis and adversarial tasks. Our findings establish a connection between architectural design, robustness, and topological properties.

## 1 Introduction

Robustness is a fundamental aspect of machine learning models (Bai et al., 2021; Wang et al., 2022). Building a robust model has various advantages, which include maintaining high performance under various input corruptions, being resilient to adversarial attack, and generalizing well to out-of-distribution data (Buzhinsky et al., 2023; Szegedy, 2013; Boopathy et al., 2019). When evaluating the robustness of models, performance analysis or intrinsic robustness analysis are mainly used (Weng et al., 2018; Wang et al., 2016; Hein & Andriushchenko, 2017). Performance analysis assesses how well a model maintains metrics like accuracy and prediction entropy under adversarial attacks, noise, or other perturbations (Carlini & Wagner, 2017; Rathnakumar et al., 2024). While this approach provides straightforward empirical evaluations, it often lacks insights into the theoretical and structural properties contributing to robustness. In contrast, intrinsic robustness analysis examines properties such as the Lipschitz continuity of models, probability distributions, or intrinsic dimensions (Szegedy, 2013; Buzhinsky et al., 2023; Tulchinskii et al., 2024). However, such techniques are either difficult to generalize (to multi-layer or other complex settings), or the given bounds are too loose. Furthermore, such methods do not allow direct theoretical robustness comparison across different architectures or configurations. Lacking theoretical foundations and relying largely on experiments for robustness analysis may lead to inconsistent conclusions and limited insights into the underlying mechanisms contributing to robustness.

To fill this gap, we propose a method for robustness comparison. By comparing the robustness of different architectures or configurations, we can gain insights into choosing or developing robust models without deriving exact robustness based on some metric. Based on the comparison method, we propose *TopoLip*, a metric based on layer-wise analysis that enables robustness comparison in both theory and experiments. Additionally, TopoLip provides insights into how model parameters influence robustness.

To introduce and validate the proposed comparison method, we use Transformer and ResNet − two dominant classifiers in vision tasks using distinct approaches − for comparison. The Transformer architecture, introduced by Vaswani (2017), has become highly popular and has made significant impacts across various fields. In contrast, ResNet, introduced by He et al. (2016), is built using convolutional layers with resid-

ual connections. As noted by Bai et al. (2021), Transformers are more robust than CNNs when handling out-of-distribution data. We further validate this by theory and experiments.

This paper is organized as follows: Section 3 presents preliminary concepts; Section 4 introduces the TopoLip robustness metric; Section 5 demonstrates that attention-based models are smoother and thus more robust than convolution-based models; and Section 6 provides experimental results to validate our theoretical findings. Our main contributions are:

- We propose a metric for evaluating the robustness of models. This metric allows for both theoretical and empirical comparisons of robustness across models and provides insights into how model parameters influence robustness.

- We establish a relationship between the Lipschitz continuity of persistence diagrams and probability distributions, linking topological data analysis to robustness evaluation.

- We analyze the mean-field regime of attention and convolution layers, comparing their Lipschitz conditions. Our findings reveal that attention layers are inherently more robust to variations in input data distributions compared to convolution layers.

- We extend the analysis to Vision Transformers (ViTs) and ResNets, demonstrating a consistent relationship between architectural design and robustness.

- Finally, we validate our theoretical insights through adversarial tasks. Experimental results confirm that attention-based models generally exhibit greater robustness than convolution-based models in handling corrupted data.

## 2 Related work

**Robustness metric.** Weng et al. (2018) converts robustness analysis into a local Lipschitz constant estimation problem to derive theoretical robustness. However, the method is algorithm-based, and the process relies on computing some metric that serves as a theoretical performance for the final robustness derivation. Therefore, it can not derive a robustness score without running the algorithm in a concrete setting, making it impossible to get insights into the model robustness completely by theory. Similarly, Weng et al. (2018) developed a robustness metric that is attack-independent and can be used with any neural network classifier. However, this approach is not well-suited for the theoretical analysis of individual models. Buzhinsky et al. (2023) proposes a metric to measure the robustness of a classifier. This metric is based on probabilistic reasoning within the latent spaces of generative models, which makes it challenging to apply to specific model settings. Hein & Andriushchenko (2017) derives a closed-form Lipschitz bound for evaluating the robustness of a multi-layer perception (MLP) with a single hidden layer. Nevertheless, a closed-form bound is hard to derive for a neural network with more than one hidden layer, not to mention increasing the complexity of the architecture such as the transformer. Wang et al. (2016) uses topology to study robustness. However, no robustness bounds or estimates were provided for neural networks, and thus no comparison can be made between architectures or model configurations.

Here, we propose a method that can not only derive a Lipschitz bound for robustness completely by theory, but the bounds of different architectures or configurations can be compared with each other by replacing the values of parameters as well. Moreover, the comparison is not only robust in theory but also in experiments.

**Topological Data Analysis.** This work is partly built upon Topological Data Analysis (TDA), which focuses on measuring the topological structures within data. The Wasserstein distance is extensively used in TDA to quantify differences between the topological structures of distributions (Cohen-Steiner et al., 2005). Although persistence diagrams (discussed in Appendix A) are not equivalent to probability spaces, they possess properties that allow for the definition of probability measures (Mileyko et al., 2011). In our study, we further explore the relationship between persistence diagrams and probability spaces, particularly in terms of their Lipschitz continuity.

## 3 Preliminaries

### 3.1 Problem setup

Suppose the input is a 2D image with resolution $(H, W)$ and $C$ channels. In Vision Transformers (ViT), the image is reshaped into a sequence of flattened patches $\mathbf{p} \in \mathbb{R}^{N \times (P^2 \cdot C)}$, where $(P, P)$ is the resolution of the patches and $N = HW/P^2$ is the number of patches (Dosovitskiy, 2020). This input is then mapped by an embedding matrix $\mathbf{E} \in \mathbb{R}^{(P^2 \cdot C) \times d}$, where $d$ is the embedding dimension. The mapping yields a matrix of size $\mathbb{R}^{N \times d}$, which can be interpreted as a sequence of $N$ input vectors $\{x_i\}_{i=1}^N \subset \mathbb{R}^d$. These vectors are often expressed as an input matrix $X = [x_1, ..., x_N] \in \mathbb{R}^{d \times N}$.

For a convolutional layer in residual networks (ResNet), let $\mathbf{y}(\alpha) \in \mathbb{R}^C$ represent the input at position $\alpha$. By utilizing a $(2k+1) \times (2k+1)$ filter, the response of a convolutional layer at position $\alpha$ can be written as $\overline{\mathbf{y}}(\alpha) = \sum_{\beta \in ker} \mathbf{W}_{::,\beta} \phi(\mathbf{y}(\alpha + \beta)) + b$, where $\mathbf{W} \in \mathbb{R}^{C \times C \times (2k+1)^2}$ is a weight matrix representing $C$ filters (where we set the $\#(filter) = \#(channel)$), $\phi$ denotes the activation function, and $b \in \mathbb{R}^C$ is a bias term. Here, for each spatial offset $\beta$ within the kernel support, $\mathbf{W}_{::,\beta} \in \mathbb{R}^{C \times C}$ is the submatrix of weights that maps the $C$-dimensional input at position $\alpha + \beta$ to a $C$-dimensional contribution in the output at position $\alpha$. Since there are $H \times W$ positions at the input image, each corresponding to one response, the input image can be regarded as a $C \times N'$ sequence where $N' = HW$. More details of the convolutional layer setting will be discussed later.

Previous works have restricted the input sequence of the attention layer $X = [x_1, ..., x_N] \in B_R^N$ where $B_R \subset \mathbb{R}^d$ is the closed ball centered at 0 and of radius $R$ (Castin et al., 2024; Geshkovski et al., 2024). We apply this restriction and assume each dimension of $x_i$ ($i \in [N]$) is drawn i.i.d. from $N(0, \sigma^2)$. Specifically, by applying Chebyshev's inequality that with high probability $1 - d/t^2$, we have $\|x_i\| \leq t\sigma$. For the convolution layer, we assume the input $Y = [y_1, ..., y_{N'}] \in B_R'^{N'}$ where $B_R' \subset \mathbb{R}^C$. Since we set $C$ infinitely large to introduce the mean-field regime of convolution, we instead bound each element: with a high probability $1 - 1/t^2$, we have $|y_{ij}| \leq t\sigma$.

### 3.2 Discrete frameworks

We define the discrete frameworks of attention and convolution same as the settings in the previous research (He et al., 2015; Chi et al., 2023).

**Definition 1** (Attention layer). *Given an input sequence $X \in \mathbb{R}^{d \times N}$, consider a single-head attention layer with parameters $\{Q_m, K_m, V_m\}_{m \in [M]} \subset \mathbb{R}^{d \times d}$. The output of the single-head attention layer is denoted as $\overline{X} = \text{Attn}_m(X) = [\overline{x}_1, \ldots, \overline{x}_N] \in \mathbb{R}^{d \times N}$, where each $\overline{x}_i$ for $i \in [N]$ is given by*

$$\overline{x}_i = \sum_{j=1}^N softmax\left(\frac{x_i^\top Q_m^\top K_m x_j}{\sqrt{d/M}}\right) V_m x_j = \sum_{j=1}^N \frac{\exp\left(x_i^\top Q_m^\top K_m x_j / \sqrt{d/M}\right)}{\sum_{k=1}^N \exp\left(x_i^\top Q_m^\top K_m x_k / \sqrt{d/M}\right)} V_m x_j. \tag{1}$$

*A multi-head attention extends this concept by allowing the model to attend to information from different representation sub-spaces jointly. A $M$-head attention layer is defined as $\text{MHAttn}(x_i, X) := \boldsymbol{o}_i$, where*

$$\boldsymbol{o}_i = W^O(\oplus_{m=1}^M head_m)$$

$$head_m = [\text{Attn}_m(X)]_{:i} = [\text{Attn}(X; \{Q_m, K_m, V_m\})]_{:i},$$

*with $W^O \in \mathbb{R}^{d \times Md}$ being learned projection matrices, and $[A]_{:i}$ denotes the $i$-th column of matrix $A$.*

Next, we define transformers. For a given input vector $x_i \in \mathbb{R}^d$, layer normalization is defined as $\text{LN}(x_i) = (x_i - \mu_i)/\sigma_i \odot \gamma + \beta$, where $\mu_i = 1/d \sum_{j=1}^d x_{i,j}$, $\sigma_i = \sqrt{\sum_{j=1}^d (x_{i,j} - \mu_i)^2/d}$, $\gamma \in \mathbb{R}^d$ and $\beta \in \mathbb{R}^d$ are learned scaling and shifting parameters, and $\odot$ denotes element-wise multiplication. An MLP layer with hidden dimension $d' = d$ is defined as $\text{MLP}(x_i) = \mathbf{W}_2 \phi(\mathbf{W}_1 x_i + \mathbf{b}_1) + \mathbf{b}_2$ where $\mathbf{W}_1, \mathbf{W}_2 \in \mathbb{R}^{d \times d}, \mathbf{b}_1, \mathbf{b}_2 \in \mathbb{R}^d$, and

$\phi$ denotes the ReLU activation. Each element in $\gamma, \beta, \mathbf{W}_1, \mathbf{W}_2, \mathbf{b}_1, \mathbf{b}_2$ is initialized following $N(0, \sigma^2)$. An $L$-layer transformer is then expressed as:

$$\text{TF}(X) = \Big( (1 + \text{MLP} \circ \text{LN}) (X + \text{MHAttn} \circ \text{LN}(X)) \Big)^L. \tag{2}$$

**Definition 2** (Convolutional layer)**.** *Consider a convolutional layer with $C$ filters and $C$ input channels. In practice, each filter could have a different size, and padding is typically applied to maintain consistent output dimensions. To ease the analysis, we set all filters have the same size $(2k + 1) \times (2k + 1)$. Let $y_i(\alpha) \in \mathbb{R}$ represents the input to the convolutional layer with filter $i$ at position $\alpha$, then the output at position $\alpha$ can be writen as*

$$\overline{y}_i(\alpha) = \sum_{c=1}^{C} \sum_{\beta \in ker} \mathtt{W}_{ci,\beta} \phi(y_c(\alpha + \beta)) + b \tag{3}$$

*where $ker := \{(p_0, p_1) \in \mathbb{Z}^2; |p_0|, |p_1| \le k\}, \mathtt{W}_{ci,\beta} \in \mathbb{R}^{C \times C}$ denotes the weight for from channel $c$ to channel $i$ at position $(\cdot + \beta)$, $b \in \mathbb{R}^C$ is the corresponding bias term, and $\phi$ is the ReLU function.*

Given a mini-batch of size $N$, and a given input sequence of vectors $X = [x_1, ..., x_N] \in \mathbb{R}^{d \times N}$, batch normalization (BN) is applied as $\text{BN}(x_i) = x_i - \mu_B / \sigma_B \odot \gamma + \beta$, where $\mu_B = 1/N \sum_{i=1}^{N} x_i$, $\sigma_B = \sqrt{1/N \sum_{i=1}^{N} (x_i - \mu_B)^2}$. An $L$-layer ResNet is then expressed as

$$\text{Res}(X) = \underbrace{(I + F) \circ \cdots \circ (I + F)}_{L \text{ times}}(X),$$

$$F(X) = \text{Conv} \circ \text{BN} \circ \text{Conv} \circ \text{BN} \circ \text{Conv} \circ \text{BN}(X). \tag{4}$$

### 3.3 Mean field frameworks

In this work, we utilize the mean-field frameworks of attention and convolution for several reasons. First, the discrete attention mechanism involves handling interactions between all pairs of elements, and it has been shown that properties such as variance depend on the input position, which makes the theoretical investigation of Lipschitz conditions challenging (Chi et al., 2023). Second, mean-field models typically exhibit smoother behavior, which is advantageous when establishing stability properties like Lipschitz continuity, as they mitigate the impact of local irregularities. Finally, the training of attention-based or convolution-based models often requires large datasets, and the mean-field framework provides a natural way to capture the macroscopic behavior of the system under such conditions.

We only define the mean-field attention layer and the mean-field convolution layer here, since our goal is to evaluate the Lipschitz continuity of models, and the Lipschitz constants of the transformer and the ResNet can be calculated by simply multiplying the Lipschitz numbers of other components.

When the input dimension $N$ is infinitely large, it can be convenient to model self-attention as a map between probability measures (Sander et al., 2022; Geshkovski et al., 2024; Castin et al., 2024). Indeed, the self-attention map is permutation equivalent. Formally, a function $f : X^N \to Y$ is said to be *permutation equivalent* if for any permutation $\pi$ of the indices $\{1, \ldots, N\}$, we have $f(x_1, \ldots, x_N) = f(x_{\pi(1)}, \ldots, x_{\pi(N)})$. This property naturally enables the map from $X = [x_1, \ldots, x_N]$ to $m(X) = \frac{1}{N} \sum_{i=1}^{N} \delta_{x_i}$, which is invariant under any permutation of the inputs.

**Definition 3** (Pushforward (Santambrogio, 2015))**.** *For a probability measure $\mu$ on $\mathbb{R}^d$ and a measurable map $\varphi : \mathbb{R}^d \to \mathbb{R}^d$, the pushforward of $\mu$ through $\varphi$, denoted as $\varphi_\# \mu$, is the probability measure defined by $(\varphi_\# \mu)(B) := \mu(\varphi^{-1}(B))$ for any Borel set $B \subset \mathbb{R}^d$, where $\varphi^{-1}(B) := \{x \in \mathbb{R}^d : \varphi(x) \in B\}$.*

**Definition 4** (Mean-field self-attention (Castin et al., 2024))**.** *Let $Q, K, V \in \mathbb{R}^{d \times d}$, and define $A := K^\top Q / \sqrt{d/M}$. Mean-field self-attention with parameters $(A, V)$ is described as:*

$$F : \mu \in \mathcal{P}_c(\mathbb{R}^d) \mapsto (\Gamma_\mu)_\# \mu, \quad \Gamma_\mu(x) = \frac{\int \exp(x^\top A^\top y) V y \, d\mu(y)}{\int \exp(x^\top A^\top y) \, d\mu(y)} \quad for \ x \in \mathbb{R}^d. \tag{5}$$

Since convolution can be permutation equivariant with respect to the channels, it can also be modeled as a map between probability measures. Specifically, the convolutional layer maps the input $Y = [y_1, ..., y_C]$ to $m'(Y) = \frac{1}{C} \sum_{c=1}^{C} \delta_{y_c}$ where $y_i(\alpha) = \sum_{\beta} \mathbf{W}_{ci,\beta} \phi(x_c(\alpha + \beta))$ is the response from channel $c$. In previous works, the number of channels is set sufficiently large to make mean field theory applicable (Xiao et al., 2018). Therefore, we can introduce the mean-field convolution based on this limit.

**Definition 5** (Mean-field convolution). *Set $\mathbf{W} \in \mathbb{R}^{C \times C \times (2k+1)^2}$. For simplicity, we denote $w_\beta \in \mathbb{R}$ as the weight from one channel to another at position $(\cdot + \beta)$. A mean-field convolutional layer with parameter $W$ is described as:*

$$G : \mu' \in \mathcal{P}_c(\mathbb{R}) \mapsto (\Gamma'_{\mu'})_{\#}\mu', \quad \Gamma'_{\mu'}(y(\alpha)) = \text{ReLU}\left( \int \sum_{\beta \in \text{ker}} w_\beta \, y(\alpha + \beta) \, d\mu'(wy) + b \right), \quad (6)$$

*where $\text{ker} = \{\beta = (\beta_1, \beta_2) \in \mathbb{Z}^2 : |\beta_1|, |\beta_2| \le k\}$ and $b \in \mathbb{R}$.*

## 4 Topological Lipschitzness

### 4.1 Robustness

We consider a classifier $f_1 = c_1 \circ g_1$ where $g_1$ maps input data from the input space $X$ to an intermediate space $X_1$, which is equipped with a distance function $d_1$. The function $c_1$ subsequently maps data from $(X_1, d_1)$ to the output space $Y$, which represents the target labels or classes. Wang et al. (2016) defines the classifier $f_1 : X \to Y$ to be $\{\delta, \eta_1\}$-robust against adversarial attacks if for any $x, x' \in X$, $\mathbb{P}(f_1(x) = f_1(x') | f_0(x) = f_0(x'), d_0(g_0(x), g_0(x')) < \delta) > 1 - \eta_1$. Here, $f_0$ represents an oracle that provides ground truth labels (e.g., a human annotator). Now, consider another classifier $f_2$ and assume the following relationships among the classifiers and their mappings:

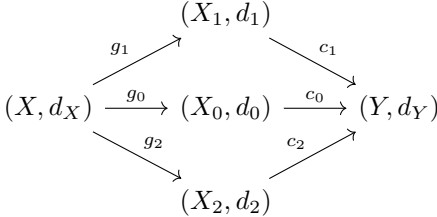

Under this framework, we can assert that $f_2$ is more robust than $f_1$ if $f_2$ is $\{\delta, \eta_2\}$-robust with the same $\delta$ but a smaller $\eta_2 < \eta_1$. Intuitively, this means that when the oracle considers two distinct inputs as belonging to the same class, $f_2$ has a higher possibility than $f_1$ of recognizing them as such under perturbations within $\delta$. In other words, if there exist inputs $x, x'$ such that $d(x, x') < \delta$ and $d_Y(f_i(x), f_i(x')) > \epsilon$ for $i = 1, 2$, then a more robust classifier satisfies $\text{Lip}_2 < \text{Lip}_1$ where $\text{Lip}_i := \max_{x,x'} d_Y(f_i(x), f_i(x'))/d(x, x')$ $(x, x' \in X)$. Therefore, by examining the Lipschitz constants of classifiers, we can gain insights into their robustness and effectively compare different models. We transform the input and output of models into probability distributions and consider the Wasserstein distance (Definition 7) in between to derive the topological Lipschitz condition.

### 4.2 Topological Lipschitzness

Before defining Topological Lipschitz continuity, we first explain why it is needed.

The Wasserstein distance is a metric between probability distributions defined on a given space. Instead of considering Lipschitz continuity between individual data points, analyzing the Lipschitz condition of the Wasserstein distance between the input and output distributions of a function provides insights into its global behavior and smoothness (Santambrogio, 2015; Villani, 2021).

However, calculating the Wasserstein distance requires that both probability distributions are defined on the same space. While one might try to use dimension reduction techniques to match distributions defined on spaces of the same dimension, such approaches are generally inefficient when computing Wasserstein distances

between layers of models like convolutional neural networks (CNNs). These networks often produce layers with different embedding spaces, and forcing them into a common space via dimension reduction can lead to information loss at different scales. Moreover, although the Gromov-Wasserstein distance is used to compare distributions on different spaces, it is computationally expensive (Zhang et al., 2024; Vayer et al., 2019) and may fail to capture subtle changes (e.g., small perturbations in feature embeddings) between distributions. This is particularly problematic when analyzing the intermediate outputs of transformer-based and residual models, where the inputs and outputs of each layer tend to be highly similar (Raghu et al., 2021) due to their architectural design.

To address this challenge, we introduce the Topological Lipschitzness (TopoLip). TopoLip is defined as the Wasserstein Lipschitz constant between the persistence diagrams (explained later) of a function's input and output. Compared to the Gromov-Wasserstein distance, computing the Wasserstein distance between persistence diagrams is both computationally efficient and sensitive to small perturbations such as stretching and squishing (Vishwanath et al., 2020). When comparing model robustness theoretically, the Wasserstein Lipschitz constants of each layer's input and output can be directly compared. Later, we will demonstrate that TopoLip is a constant multiple of the Wasserstein Lipschitz constant. Experimentally, the input and output are transformed into persistence diagrams, and their Wasserstein distance is computed. The maximum rate of change of this Wasserstein distance defines TopoLip.

Persistence diagrams are a tool from topological data analysis (TDA) used to summarize the underlying structure of data. They track significant features—such as connected components, loops, or voids—that emerge and vanish as the scale of observation varies. In a persistence diagram, each feature is represented by a point whose coordinates indicate the scale at which the feature appears (its "birth") and the scale at which it disappears (its "death"). For further details on persistence diagrams, see Appendix A.

Informally, TopoLip measures the Lipschitz condition of the Wasserstein distance between the persistence diagrams of a function's input and output. The relationship can be illustrated as follows:

$$\text{Input Distribution} \xrightarrow{\ F\ } \text{Feature Embeddings} \xrightarrow{\ g\ } \text{Persistence Diagrams}$$

$$\text{(Probability Distribution)}$$

Here, TopoLip combines the Lipschitz constant of the function $F$ and that of the Lipschitz map $g$ that generates persistence diagrams. Formally, by Lemma 1, TopoLip is defined as follows:

**Definition 6.** *Let $g$ be a Lipschitz map defined by:*

$$g : \ \mathcal{D} \ \longrightarrow \ \mathcal{PD}_k,$$

$$g(X) = \{(b_i, d_i) \mid \text{feature } i \text{ in } H_k(X) \text{ is born at } b_i \text{ and dies at } d_i\},$$

*where $\mathcal{D}$ is the space of finite metric spaces (datasets), $H_k(X)$ denotes the $k$-th homology group of $X$ (which captures $k$-dimensional topological features), and $\mathcal{PD}_k$ is the space of persistence diagrams for dimension $k$ endowed with the Wasserstein distance $\mathcal{W}_p$ ($p \geq 1$). For a Lipschitz function $F$, its Topological Lipschitzness is defined as:*

$$\text{Lip}_{\text{TopoLip}}^{\mathcal{W}_p}(F) := \text{Lip}^{\mathcal{W}_p}(g) \cdot \text{Lip}^{\mathcal{W}_p}(F).$$

The map $g$ is Lipschitz due to the stability theorem presented in Cohen-Steiner et al. (2005). When $g$ is fixed (in this work, the Čech complex filtration) to generate persistence diagrams, $\text{Lip}(g)$ remains constant. Therefore, the TopoLip of a function is directly proportional to its Wasserstein Lipschitz constant. By examining the Wasserstein Lipschitz constant of a model, we can compare the TopoLip of different models and thus assess their robustness.

## 5 Wasserstein Lipschitzness comparison

We begin by defining the Wasserstein Lipschitzness:

**Definition 7** (Lipschitz constant with respect to the $\infty$-Wasserstein distance). *Denote $\mathcal{P}_c(\mathbb{R}^d)$ the set of compactly supported probability measures on $\mathbb{R}^d$. For $\mu, \nu \in \mathcal{P}_c(\mathbb{R}^d)$, the $\infty$-Wasserstein distance, also known as the bottleneck distance, is defined as:*

$$\mathcal{W}_\infty(\mu, \nu) := \inf_{\pi \in \Pi(\mu,\nu)} \sup_{(x,y) \in supp(\pi)} \|x - y\|,$$

*where $\Pi(\mu, \nu)$ is the set of couplings between $\mu$ and $\nu$. For a map $F : \mathcal{P}_c(\mathbb{R}^d) \to \mathcal{P}_c(\mathbb{R}^d)$ and any subset $\mathcal{X} \subset \mathcal{P}_c(\mathbb{R}^d)$, the Lipschitz constant of $F$ on $\mathcal{X}$ is defined as:*

$$\mathrm{Lip}^{\mathcal{W}_\infty}(F_{|\mathcal{X}}) := \sup_{\mu,\nu \in \mathcal{X}, \mu \neq \nu} \frac{\mathcal{W}_\infty(F(\mu), F(\nu))}{\mathcal{W}_\infty(\mu, \nu)}.$$

*If $\mathrm{Lip}^{\mathcal{W}_\infty}(F_{|\mathcal{X}})$ is finite, then $F$ is said to be $\mathcal{W}_\infty$-Lipschitz continuous on $\mathcal{P}_c(\mathbb{R}^d)$.*

The reason for using $\mathrm{Lip}^{\mathcal{W}_\infty}$ instead of $\mathrm{Lip}^{\mathcal{W}_1}$ or $\mathrm{Lip}^{\mathcal{W}_2}$ is that for any pair of probability measures $\mu$ and $\nu$, $\mathcal{W}_1(\mu, \nu) \leq \mathcal{W}_2(\mu, \nu) \leq \mathcal{W}_\infty(\mu, \nu)$ holds. This inequality indicates that the $\mathcal{W}_\infty$ metric captures the maximum (or worst-case) discrepancy between the measures. Consequently, a Lipschitz condition defined in terms of $\mathcal{W}_\infty$ ensures that the mapping is stable with respect to the largest deviation between $\mu$ and $\nu$, making it a tighter and more stringent requirement than those based on $\mathcal{W}_1$ or $\mathcal{W}_2$.

To ensure a fair comparison of variances between the self-attention and convolutional layers, we take each element of $Q, K, V, W^O$ in the self-attention layer to be drawn i.i.d. from $\mathcal{N}(0, \sigma^2)$. For the convolution layer, to follow common initialization schemes such as He initialization (He et al., 2015), each element of $W$ is drawn from i.i.d. $\mathcal{N}(0, \sigma^2/(C(2k+1)^2))$. We assume $H, W, C$ in the input image size $H \times W \times C$ are very large. For the self-attention layer, the input is a sequence with size $d \times N$, where $d$ is the embedding dimension and $N = HW/P^2$. For the convolution layer, the input is a sequence with size $C \times N'$ where $N' = HW$.

To apply the mean-field framework, the discrepancy between its outputs and those of the discrete models must be negligible. From the Central Limit Theorem, averaging over $n$ independent samples results in a standard deviation proportional to $1/\sqrt{n}$. For a convolutional layer, the sum over $C$ channels has fluctuations of order $1/\sqrt{C}$, and for a self-attention layer, the sum over $N$ tokens yields fluctuations of order $1/\sqrt{N}$. To ensure the mean-field approximation holds at a given fluctuation level—such as no more than 5%—we require $N, C \gtrsim 400$. Such values are commonly found in practice—for instance, in $28 \times 28$ images or ResNet layers with 512 filters.

## 5.1 Attention and convolution

**Theorem 1.** *Let $Q, K, V \in \mathbb{R}^{d \times d}$. For any $t > \sqrt{d}$ and $s \geq \sigma\sqrt{2\log 2}$, with probability at least $\min\{1 - d/t^2, 1 - 2e^{-s^2/(2\sigma^2)}\}$, and assuming the operator norm $\|A\|_{op} \geq 2/\sigma^2$, the mean-field single-head attention map $\mathrm{Attn}_{|\mathcal{P}(B_{t\sigma})}$ with parameter $(Q, K, V)$ is $\mathcal{W}_\infty$-Lipschitz continuous on the set $\mathcal{P}(B_{t\sigma})$, and its Lipschitz constant is bounded by*

$$\mathrm{Lip}^{\mathcal{W}_\infty}(\mathrm{Attn}_{|\mathcal{P}(B_{t\sigma})}) = 2t\sigma(2\sigma\sqrt{d} + s)^2(1 + t\sigma d^{-1/2}(2\sigma\sqrt{d} + s)^2)$$

*Similarly, the Lipschitz constant of mean-field $M$-head attention map $\mathrm{MHAttn}_{|\mathcal{P}(B_{t\sigma})}$ is bounded by*

$$\mathrm{Lip}^{\mathcal{W}_\infty}(\mathrm{MHAttn}_{|\mathcal{P}(B_{t\sigma})}) = 2t\sigma\sqrt{M}(2\sigma\sqrt{d} + s)^2(1 + t\sigma\sqrt{\frac{M}{d}}(2\sigma\sqrt{d} + s)^2).$$

The choice of using the mean-field attention model instead of the discrete attention mechanism is motivated by several considerations:

To simplify the upper bounds, assume $t = p\sqrt{d}$, $s = q\sigma$ for constants $p, q > 0$. Under this assumption, the Lipschitz constants of a single-head and multi-head attention layer can be approximated as follows:

$$\mathrm{Lip}^{\mathcal{W}_\infty}(\mathrm{Attn}_{|\mathcal{P}(B_{t\sigma})}) = \mathcal{O}(\sigma^6 d^{5/2}), \ \mathrm{Lip}^{\mathcal{W}_\infty}(\mathrm{MHAttn}_{|\mathcal{P}(B_{t\sigma})}) = \mathcal{O}(\sigma^6 d^{5/2} M).$$

**Theorem 2.** *Let* $\mathtt{W} \in \mathbb{R}^{C \times C \times (2k+1)^2}$ *where* $\mathtt{W}_{ci,\beta} \sim N(0, \frac{\sigma^2}{C(2k+1)^2})$ *represents the weight from channel $c$ to channel $i$ at position $(\cdot + \beta)$. Denote the output vector of the mean-field convolutional layer as* $\overline{\mathbf{y}}(\alpha) = \left[\overline{y}_1(\alpha), \cdots, \overline{y}_C(\alpha)\right]$ *where* $\overline{y}_i(\alpha) = \int_{\mathbb{R}} \left(\sum_{\beta} \mathtt{W}_{ci,\beta} y_i(\alpha + \beta) + b_i\right) d\mu(\mathtt{W}y)$. *For any $t > 0$, with probability at least $1 - 1/t^2$, the Lipschitz constant of the mean-field convolution map* $\mathrm{Conv}_{|\mathcal{P}(B_{t\sigma})}$ *with parameter $\mathtt{W}$ is bounded by*

$$\mathrm{Lip}^{\mathcal{W}_\infty}(\mathrm{Conv}_{|\mathcal{P}(B_{t\sigma})}) = (2k+1)\sqrt{t\sigma C\left(1 + \frac{1}{(2k+1)\sqrt{C}}\right)} = \mathcal{O}(k\sqrt{\sigma C}).$$

*where we assume $t$ to be some moderate positive number to simplify the upper bound.*

The proofs of Theorem 1 and 2 are available in Appendix B.

From the above bounds, we observe that the Wasserstein Lipschitz constant of attention layers, as well as their TopoLip and robustness, are highly related to the embedding dimension $d$ and the head number $M$. Since $d$ and $M$ are fixed, we can indicate that $\mathrm{Lip}^{\mathcal{W}_\infty}$ of attention layers remains in a certain range. For convolution layers, since their Wasserstein Lipschitz constant is related to the channel number $C$ which usually is not fixed in a model, its robustness tends to be lower than attention layers.

Suppose the bounds in Theorem 1 and 2 are tight, then we can assess the Lipschitz bounds of both models from a practical perspective. In practice, typical parameter values are often set as follows: $\sigma \sim 10^{-2}$, $d \sim 10^2$, $M \sim 10^1$, $k \sim 10^1$, and $C \sim 10^2$. Under this setting, the Lipschitz bound for multi-head attention is on the order of $\mathcal{O}(10^{-6})$, whereas that for convolutional layers is significantly larger, around $\mathcal{O}(10^1)$. To provide a more concrete comparison, consider the following specific parameter settings: $d = 512$, $M = 8$, $\sigma = 0.05$, $k = 3$, and $C = 512$. Under this setting, $\sigma^5 d^2 \approx 0.08$, $\sigma^6 d^{5/2} M \approx 0.74$, while $k\sqrt{\sigma C} \approx 15$. Furthermore, it is important to note that $C$ is not fixed in practice. For instance, the number of channels in ResNet50 are 64→256→512→1024→2048, which leads to a larger Lipschitz bound for convolutional layers. Therefore, convolution is more unstable under this setting, leading to greater TopoLip and lower robustness.

Theorems 1 and 2 indicate that while the $\mathrm{Lip}^{\mathcal{W}_\infty}$ bound for convolution depends on $C$, which can vary across different convolutional models, the $\mathrm{Lip}^{\mathcal{W}_\infty}$ bound for attention remains fixed and is relatively tight under practical settings. In a real-life scenario, attention and convolution layers are rarely used solely. Instead, they are one part of the models. To conduct a thorough comparison, we extend our investigation to two widely used models: Vision Transformer (ViT) and residual neural network (ResNet).

## 5.2 ViT and ResNet

We consider the Vision Transformers (ViTs) and ResNets. Building upon the calculations presented in Theorems 1 and 2, and utilizing Lemma 1, we have the following theorems.

**Theorem 3.** *For any $t > \sqrt{d}$ and $s \geq \sigma\sqrt{2\log 2}$, with probability at least $\min\{1 - d/t^2, 1 - 2e^{-s^2/(2\sigma^2)}\}$, and assuming $A$ in Definition 5 satisfies $\|A\|_{op} \leq 2/\sigma^2$, the transformer defined in Equation 2 is $\mathcal{W}_\infty$-Lipschitz continuous on the set $\mathcal{P}(B_{t\sigma})$, and its Lipschitz constant is bound by*

$$\mathrm{Lip}^{\mathcal{W}_\infty}(\mathrm{TF}_{|\mathcal{P}(B_{t\sigma})}) \leq \left(1 + t(2\sigma\sqrt{d} + s)^2\right)^L \left(1 + 2t^2\sigma\sqrt{M}(2\sigma\sqrt{d} + s)^2(1 + t\sigma\sqrt{\frac{M}{d}}(2\sigma\sqrt{d} + s)^2)\right)^L. \quad (7)$$

Using the same approach above that assuming $t = p\sqrt{d}$, $s = q\sigma$ for constants $p, q > 0$, the Lipschitz constants becomes

$$\mathrm{Lip}^{\mathcal{W}_\infty}(\mathrm{TF}_{|\mathcal{P}(B_{t\sigma})}) = \mathcal{O}\left(\max\left\{1, \sigma^2 d^{3/2}, \sigma^6 d^3 M, \sigma^8 d^{9/2} M\right\}\right)^L. \quad (8)$$

**Theorem 4.** *For any $t > 0$, with probability at least $1 - 1/t^2$, the ResNet defined in Equation 4 is $\mathcal{W}_\infty$-Lipschitz continuous on the set $\mathcal{P}(B_{t\sigma})$, and its Lipschitz constant is bound by*

$$\mathrm{Lip}^{\mathcal{W}_\infty}(\mathrm{Res}_{|\mathcal{P}(B_{t\sigma})}) \leq \left(1 + t^{9/2}(2k+1)^3\left(\sigma C\left(1 + \frac{1}{(2k+1)\sqrt{C}}\right)\right)\right)^{3/2}\right)^L = \mathcal{O}\left(\max\left\{1, t^{9/2}k^3\sigma^{3/2}C^{3/2}\right\}\right)^L. \quad (9)$$

Table 1: Model Comparisons.

| Parameter Impact | Configuration |
|---|---|
| Depth (ResNet) | Kernel size 3, ResNet18/50/101/152 |
| Kernel size (ResNet) | ResNet18, kernel size 3/7/11 |
| Depth (ViT) | Head 12, embedding dimension 768, depth 6/12/24 |
| Head (ViT) | Depth 12, embedding dimension 768, head 6/12/16 |
| Embedding dimension (ViT) | Head 12, depth 12, embedding dimension 384/768/1024 |

The proofs of Theorem 3 and 4 are available in Appendix B.

From the results, we observe that the Lipschitz constants $\text{Lip}^{\mathcal{W}_\infty}$ for both ViTs and ResNets retain and further magnify the parameter dependencies inherent in their respective attention and convolutional layers. Specifically, under the same conditions as those discussed in Section 5.1, we find that $\text{Lip}^{\mathcal{W}_\infty}(\text{TF}) = \mathcal{O}(10^{-1})$ for ViTs, while $\text{Lip}^{\mathcal{W}_\infty}(\text{Res}) = \mathcal{O}(t^{9/2})$ for ResNets. Given that Theorem 4 holds with probability $1 - 1/t^2$ for $t > 1$, $t$ cannot be too small. As a result, $\text{Lip}^{\mathcal{W}_\infty}(\text{Res})$ is generally larger than $\text{Lip}^{\mathcal{W}_\infty}(\text{TF})$. Additionally, since the number of channels $C$ in ResNet can be very large, the Lipschitz constant for ResNet can become significantly higher than that of ViT. As a result, ViTs tend to have a lower TopoLip value, which means they are smoother in terms of their topological properties compared to ResNets. This smoothness suggests that ViTs are less affected by changes or noise in the input, which could make them more stable and robust in their performance.

# 6 Experimental results

We conduct experiments on the CIFAR-10 and CIFAR-10-C datasets to evaluate the relationship between TopoLip and model robustness (Hendrycks & Dietterich, 2019). Specifically, we train various ResNets (ResNet18/50/101/152) and seven ViTs under practical settings.

For ResNets, we train ResNet18/50/101/152 with a kernel size of 3 to examine the effect of network depth, and train ResNet18 with kernel sizes of 3/7/11 to investigate the impact of kernel size on TopoLip and robustness (Table 1). Each model is trained for 150 epochs on CIFAR-10 until performance plateaued. The evaluation accuracy for ResNets ranges from 86.72% to 90.61% (Figure 10 and 11). For ViTs, we use models with head 12, embedding dimension 768, and depths of 6/12/24 to study the effect of depth; models with embedding dimension 768, depth 12, and heads of 6/12/16 to evaluate the effect of the number of heads; and models with head 12, depth 12, and embedding dimensions of 384/768/1024 to examine the impact of embedding dimension (Table 1). Each ViT model is trained for 200 epochs on CIFAR-10 until performance plateaued. The evaluation accuracy for ViTs ranges from 78.39% to 80.32% (Figure 12, 13, and 14). Finally, we compare TopoLip and robustness between ResNets and ViTs to verify our theoretical predictions.

After training, we first evaluate the TopoLip of the models and investigate its relationship with robustness. To measure the Bottleneck distance between the persistence diagrams of the input and output at each layer (or each block for ResNets), we switch the models to evaluation mode to freeze their parameters. We then input the test dataset and collect the outputs from all layers. Using these outputs, we compute their persistence diagrams and calculate the Bottleneck distances between adjacent layers. Next, we compute the absolute change rates between consecutive layers. Given two adjacent layers with Bottleneck distances $\mathcal{W}_{\infty,1}$ and $\mathcal{W}_{\infty,2}$, the absolute change rate between them is defined as $|(\mathcal{W}_{\infty,2} - \mathcal{W}_{\infty,1})/\mathcal{W}_{\infty,1}|$. Finally, the TopoLip is computed as the **maximum absolute change rate** observed across all layers. Here, we compare the TopoLips of ResNet18/50/101/152 and ViTs with varying depths (Table 2). From Table 2, we observe that as model depth increases, TopoLip values increase for both ResNets and ViTs. Furthermore, ViTs with fewer layers (6 or 12) exhibit lower TopoLip values compared to ResNets, whereas the TopoLip of ViT($L = 24$) exceeds that of all ResNets.

Next, we evaluate model robustness using CIFAR-10-C and analyze the relationship between TopoLip and model robustness. Specifically, we focus on three corruption types: "Gaussian noise," "impulse noise," and

Table 2: TopoLip on CIFAR-10. Here, "Res" denotes ResNet, and $L$ represents the depth of ViT with embedding dimension 768 and 12 heads.

| | **Res18** | **Res50** | **Res101** | **Res152** | **ViT**($L = 6$) | **ViT**($L = 12$) | **ViT**($L = 24$) |
|---|---|---|---|---|---|---|---|
| **TopoLip** ($\downarrow$) | 0.43 | 0.57 | 0.78 | 1.17 | 0.39 | 0.70 | 1.56 |

"pixelate." Since CIFAR-10-C includes five corruption levels for each type, we compute the model accuracy at each level and take the average. We then calculate the mean Corruption Error (mCE) and the Expected Calibration Error (ECE) based on CIFAR-10-C results. Beyond corruption robustness, we also examine adversarial and perturbation robustness. For adversarial robustness, we generate adversarial examples using the Fast Gradient Sign Method (FGSM), perturbing images along the gradient direction with strength $\epsilon = 0.01$, and measure model accuracy under these attacks. For perturbation robustness, we introduce Gaussian noise with a standard deviation of $\sigma = 0.01$ to the images and evaluate model performance on the noisy inputs. Since ViTs and ResNets achieve similar model accuracies, we assess their robustness by measuring the percentage drop in accuracy when under attack.

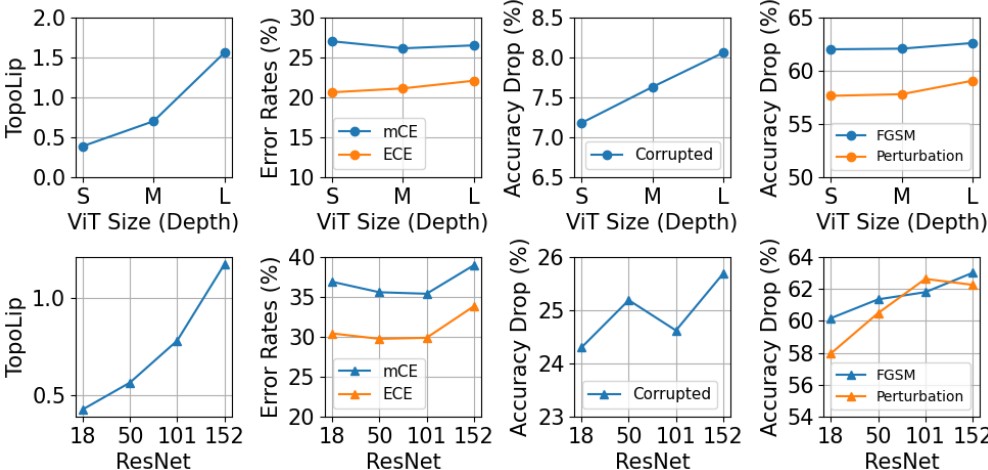

Figure 1: TopoLip and robustness performance of ViTs and ResNets under Gaussian noise corruption. Evaluated models include ResNet18/50/101/152 and ViTs with 12 attention heads, 768 embedding dimensions, and 6 (S), 12 (M), or 24 (L) layers. A higher ECE, mCE, or accuracy drop suggests lower robustness.

The results of ResNets and ViTs with varying depths under Gaussian noise corruption are shown in Figure 1. From the figure, we observe that models with higher ECE, mCE, or accuracy drop tend to have higher TopoLip values. Although both ResNets and ViTs exhibit this trend, it is less pronounced for smaller models. For example, ResNet50 and ResNet101 show similar robustness, as do ViT($L = 6$) and ViT($L = 12$).

Next, we examine the impact of other parameters on model robustness. Figure 2 shows that ViTs with higher embedding dimensions tend to suffer a larger accuracy drop under corruption, even though their mCE and ECE are not necessarily lower. Furthermore, the number of attention heads has little effect on both robustness and TopoLip. This can be explained by the fact that the Lipschitz constant of ViTs depends on $M$ at first order, but on $d$ and $L$ at higher orders. As a result, changes in $M$ have limited impact on the Lipschitz condition, while $d$ and $L$ have a much larger effect. Consequently, the overall Lipschitz condition of the model is only slightly influenced, especially considering that the calculated Lipschitz constant is not necessarily tight.

Then, we investigate the impact of kernel size on the robustness of ResNets. From Figure 3, we observe that a larger kernel size results in a higher TopoLip and a more robust model. The calculated Lipschitz constant indicates that since TopoLip depends on multi-order $k$, it increases as $k$ grows. However, a larger $k$ also enhances the robustness of ResNets. This is because a larger $k$ leads to a greater receptive field,

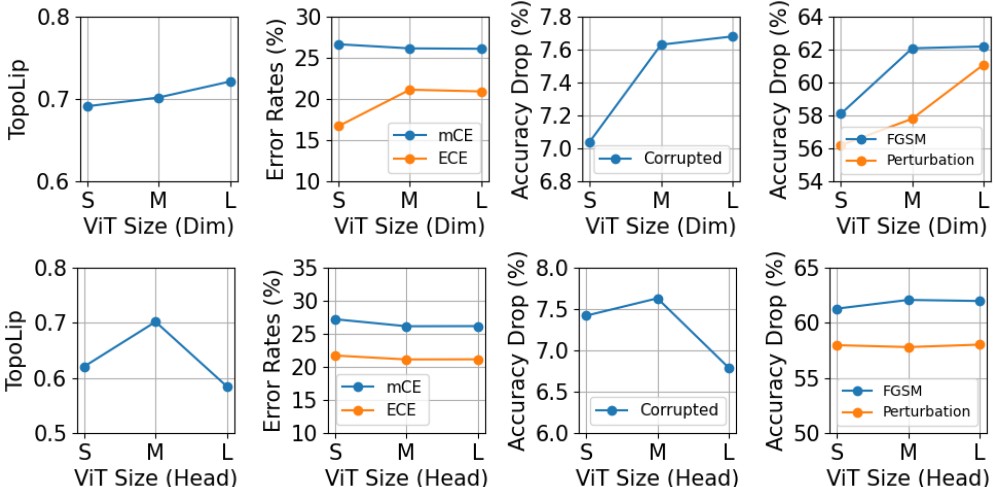

Figure 2: TopoLip and robustness performance of ViTs under Gaussian noise corruption. Evaluated models include ViTs with 12 layers and 12 heads at embedding dimensions 384 (S), 768 (M), and 1024 (L), as well as ViTs with 12 layers, 768 embedding dimensions, and 6 (S), 12 (M), or 16 (L) heads. A higher mCE, ECE, or accuracy drop suggests lower robustness.

allowing ResNet to smooth out local noise and small adversarial perturbations by averaging the response over a wider area. Consequently, the network becomes less sensitive to minor, localized changes, thereby improving robustness.

Additionally, training results indicate that ResNets with larger kernel sizes are more prone to overfitting, as larger kernels contain more parameters (Figure 11). Similar trends are observed in ViTs, where larger models with more parameters are more susceptible to overfitting (Figures 12, 13, and 14).

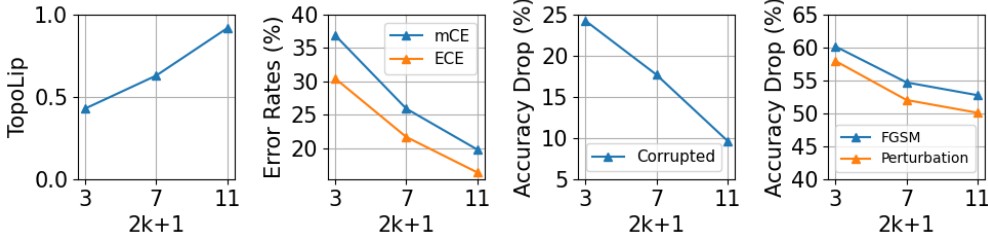

Figure 3: TopoLip and robustness of ResNet18 models with kernel sizes 3, 7, and 11 under Gaussian noise corruption. A higher mCE, ECE, or accuracy drop indicates lower robustness.

Results under impulse corruption and pixelate corruption are shown in Appendix C.1.

## 7  Conclusion

In this paper, we introduced *TopoLip*, a metric designed to assess the robustness of machine learning models at a layer-wise level. TopoLip can be used for both theoretical and experimental comparisons of different architectures or configurations, and can provide insights into how model robustness depends on parameters. Through theoretical analysis of the Wasserstein-Lipschitz conditions in mean-field attention and convolution, we revealed that attention-based models are inherently smoother than convolutional models, making them more robust as defined by TopoLip. Experimental results further validated these findings, showing that attention-based models exhibit greater robustness than convolution-based models when handling corrupted data.

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

# A    Persistence Homology and Topological Data Analysis

Topological Data Analysis (TDA) offers a set of tools from algebraic topology to study the shape of data. One central concept in TDA is *persistent homology*, which captures the evolution of topological features across multiple scales. See Le & Yamada (2018); Bubenik et al. (2015); Naitzat et al. (2020) for more details.

The process begins with the construction of a *filtration*, which is a one-parameter family of nested simplicial complexes:

$$\emptyset = K_0 \subseteq K_{r_1} \subseteq K_{r_2} \subseteq \cdots \subseteq K_{r_n} = K,$$

where $0 \leq r_1 < r_2 < \cdots < r_n$ are scale parameters. A common way to build such a filtration is via the Čech complex. Given a finite set of points $X = \{x_1, \ldots, x_N\}$ in a metric space, the Čech complex at scale $r$ is defined as

$$K_r = \Big\{ \sigma \subseteq X \;\Big|\; \bigcap_{x \in \sigma} B(x, r) \neq \emptyset \Big\},$$

where $B(x, r)$ denotes the closed ball of radius $r$ centered at $x$. As the radius $r$ increases, more simplices are added (see Figure 4), capturing the topology of the data at different scales.

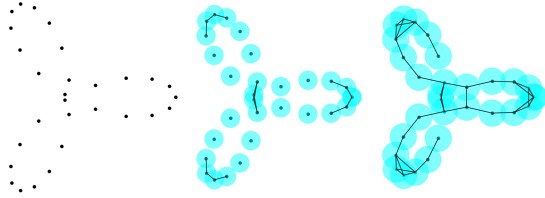

Figure 4: Construction of the Čech complex: Balls of radius $r$ are centered at data points, and simplices are formed when these balls intersect.

As the filtration progresses, topological features appear (are born) and disappear (die). These features are summarized by the homology groups $H_i$: $H_0$ records connected components, $H_1$ captures loops (or one-dimensional holes), and higher homology groups $H_i$ describe $i$-dimensional voids. Persistent homology tracks the birth and death scales of these features, assigning each a lifespan $[b, d]$. This information is commonly visualized as a *persistence barcode* (left of Figure 5) or, equivalently, as a *persistence diagram* (right of Figure 5), where each point $(b, d)$ represents a feature born at $b$ and dying at $d$.

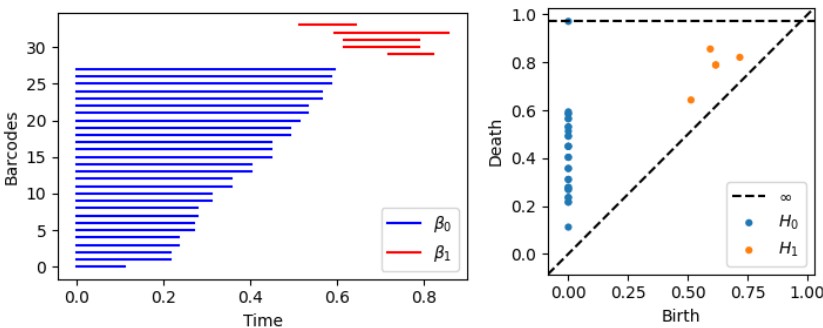

Figure 5: Left: Persistence barcode, where each bar represents the lifespan of a topological feature. Right: Persistence diagram, with each point $(b, d)$ corresponding to a feature born at $b$ and dying at $d$.

Mathematically, a persistence diagram can be viewed as a discrete measure $\mu_{\mathcal{D}_g} = \sum_{u \in \mathcal{D}_g} \delta_u$, where $\delta_u$ is the Dirac measure at $u \in \mathbb{R}^2$. To compare the topological signatures of different datasets, metrics such as the *bottleneck distance* and the *p-Wasserstein distances* are employed (Le & Yamada, 2018; Adams et al., 2017; Berwald et al., 2018).

## B  Proof of Section 5

**Lemma 1** ((Lipschitz Constant of Composed Functions (Gouk et al., 2021))). *Let $(X, d_X)$, $(Y, d_Y)$, and $(Z, d_Z)$ be metric spaces. Suppose that $f : X \to Y$ is Lipschitz continuous with Lipschitz constant $L_f$, and $g : Y \to Z$ is Lipschitz continuous with Lipschitz constant $L_g$. Then the composition $g \circ f : X \to Z$ is Lipschitz continuous with Lipschitz constant at most $L_f \cdot L_g$. In other words, for all $x_1, x_2 \in X$,*

$$d_Z(g(f(x_1)), g(f(x_2))) \leq L_f \cdot L_g \cdot d_X(x_1, x_2).$$

**Lemma 2** ((Vershynin, 2010)). *Given a matrix $A \in \mathbb{R}^{d \times d}$ with entries $A_{ij} \sim_{i.i.d.} N(0, \sigma^2)$, denote the singular values as $s_1(A) \geq s_2(A) \geq \cdots \geq s_d(A) \geq 0$. Then:*

$$P[s_1(A) \leq 2\sigma\sqrt{d} + t] \geq 1 - 2e^{-\frac{t^2}{2\sigma^2}}.$$

**Proof of Theorem 1.** We begin by bounding the Lipschitz constant for single-head attention. While Castin et al. (2024) provides an upper bound for $\mathrm{Lip}(\mathrm{Attn}_{|\mathcal{P}(B_{t\sigma})})$, their proof is abbreviated. Here, we present the comprehensive proof and offer a potentially tighter lower bound. We also extend the analysis to multi-head attention by providing an upper bound for $\mathrm{Lip}(\mathrm{MHAttn}_{|\mathcal{P}(B_{t\sigma})})$.

Define the kernel function $K(x, y) := \exp(x^\top A^\top y)$. The mean-field attention map is then expressed as:

$$\Gamma_\mu(x) = \int_{\mathbb{R}^d} \frac{K(x, y)Vy}{\int K(x, y)d\mu(y)} d\mu(y).$$

To bound the Lipschitz constant, we consider the difference between $\Gamma_\mu$ and $\Gamma_\nu$ for two probability measures $\mu$ and $\nu$ in $\mathcal{P}(B_{t\sigma})$:

$$\|\Gamma_\mu(x) - \Gamma_\nu(x)\|_{L^\infty(B_{t\sigma}, \mathbb{R}^d)}$$
$$= \left| \frac{\int_{\mathbb{R}^d} K(x, y)Vy d\mu(y) \int_{\mathbb{R}^d} K(x, y)d\nu(y) - \int_{\mathbb{R}^d} K(x, y)Vy d\nu(y) \int_{\mathbb{R}^d} K(x, y)d\mu(y)}{\int_{\mathbb{R}^d} K(x, y)d\mu(y) \int_{\mathbb{R}^d} K(x, y)d\nu(y)} \right|.$$

Denote $y^* := \max_{y \in B_{t\sigma}} \|y\|$. We bound the numerator first:

$$\left| \int_{\mathbb{R}^d} K(x, y)Vy d\mu(y) \int_{\mathbb{R}^d} K(x, y)d\nu(y) - \int_{\mathbb{R}^d} K(x, y)Vy d\nu(y) \int_{\mathbb{R}^d} K(x, y)d\mu(y) \right|$$
$$= \left| \int_{\mathbb{R}^d} K(x, y)Vy d\mu(y) \int_{\mathbb{R}^d} K(x, y)(d\nu - d\mu)(y) \right.$$
$$\left. - \int_{\mathbb{R}^d} K(x, y)Vy(d\nu - d\mu)(y) \int_{\mathbb{R}^d} K(x, y)d\mu(y) \right|$$
$$\leq \left| \int_{\mathbb{R}^d} K(x, y)d\mu(y) \right| \left( \|V\|_{op} y^* \left| \int_{\mathbb{R}^d} K(x, y)(d\nu - d\mu)(y) \right| + \left| \int_{\mathbb{R}^d} K(x, y)Vy(d\nu - d\mu)(y) \right| \right)$$
$$\leq 2\|V\|_{op} y^* \left| \int_{\mathbb{R}^d} K(x, y)d\mu(y) \right| \left| \int_{\mathbb{R}^d} K(x, y)(d\nu - d\mu)(y) \right|$$
$$\leq 2\|V\|_{op} y^* \left| \int_{\mathbb{R}^d} K(x, y)d\mu(y) \right| \|K(x, \cdot)\|_{C^{0,1}(B_{t\sigma})} \mathcal{W}_1(\mu, \nu)$$
$$\leq 2y^* \|V\|_{op} \left| \int_{\mathbb{R}^d} K(x, y)d\mu(y) \right| \|K(x, \cdot)\|_{C^{0,1}(B_{t\sigma})} \mathcal{W}_\infty(\mu, \nu)$$

where we use the inequality $\mathcal{W}_1(\mu, \nu) \leq \mathcal{W}_\infty(\mu, \nu)$. By Lemma 2, with probability at least $1 - 2e^{-s^2/(2\sigma^2)}$, we have $\|V\|_{op} \leq 2\sigma\sqrt{d} + s$, $\|A\|_{op} \leq \sqrt{\frac{M}{d}}\|K\|_{op}\|Q\|_{op} \leq \sqrt{\frac{M}{d}}(2\sigma\sqrt{d} + s)^2$, where $\|\cdot\|_{op}$ is the operator norm. For $\|K(x, \cdot)\|_{C^{0,1}(B_{t\sigma})}$, we can bound it as follows:

$$\|K(x, \cdot)\|_{C^{0,1}(B_{t\sigma})}$$

$$
= \sup_{y \in B_{t\sigma}} |K(x,y)| + \sup_{y_1 \neq y_2 \in B(0, t\sigma)} \frac{|K(x, y_1) - K(x, y_2)|}{\|y_1 - y_2\|}
$$

$$
\leq \sup_{y \in B_{t\sigma}} |K(x,y)| + \sup_{y \in B_{t\sigma}} \|\nabla_y K(x,y)\|
$$

$$
\leq K^*(x,y) + y^* \|A\|_{op} K^*(x,y)
$$

$$
= K^*(x,y)(1 + y^* \|A\|_{op})
$$

where $K^*(x,y) := \sup_{y \in B_{t\sigma}} K(x,y) = \exp(y^* \|x^\top A\|)$ and the first inequality follows from the definition of the $C^{0,1}$ norm and the mean value theorem. Then $\|\Gamma_\mu(x) - \Gamma_\nu(x)\|_{L^\infty(B_{t\sigma}, \mathbb{R}^d)}$ can be bounded by

$$
\|\Gamma_\mu(x) - \Gamma_\nu(x)\|_{L^\infty(B_{t\sigma}, \mathbb{R}^d)}
$$

$$
\leq \frac{2y^* \|V\|_{op} \left| \int_{\mathbb{R}^d} K(x,y) d\mu(y) \right| K^*(x,y)(1 + y^* \|A\|_{op})}{\left| \int_{\mathbb{R}^d} K(x,y) d\mu(y) \int_{\mathbb{R}^d} K(x,y) d\nu(y) \right|} \mathcal{W}_\infty(\mu, \nu)
$$

$$
= 2y^* \|V\|_{op} (1 + y^* \|A\|_{op}) \frac{K^*(x,y)}{\int_{\mathbb{R}^d} K(x,y) d\nu(y)} \mathcal{W}_\infty(\mu, \nu).
$$

To bound the integral part, we transform $\int d\nu(y)$ to $\int p(y) dy$ where $p(y)$ is the probability density function (pdf) of $y$. Since $y \sim N(0, \sigma^2 I)$, by using the pdf of the multivariate Gaussian distribution, we have

$$
\int_{R^d} K(x,y) d\nu(y) = \int_{R^d} K(x,y) p(y) dy
$$

$$
= \frac{1}{(2\pi\sigma^2)^{d/2}} \int_{R^d} e^{x^\top A y} \cdot e^{-\|y\|^2/(2\sigma^2)} dy
$$

$$
= e^{\sigma^2 \|x^\top A\|^2/2} \frac{1}{(2\pi\sigma^2)^{d/2}} \int_{R^d} e^{-\|y - \sigma^2 x^\top A\|^2/(2\sigma^2)} dy
$$

$$
= e^{\sigma^2 \|x^\top A\|^2/2}.
$$

Therefore,

$$
\frac{K^*(x,y)}{\int_{\mathbb{R}^d} K(x,y) d\nu(y)} = \exp(y^* \|x^\top A\| - \sigma^2 \|x^\top A\|^2/2).
$$

To bound it at 1, we need to ensure that

$$
y^* \leq \frac{\sigma^2}{2} \|x^\top A\| \leq \frac{y^* \sigma^2}{2} \|A\|_{op} \implies \|A\|_{op} \geq \frac{2}{\sigma^2}.
$$

holds. Under this condition, the final bound is

$$
\|\Gamma_\mu(x) - \Gamma_\nu(x)\|_{L^\infty(B_{t\sigma}, \mathbb{R}^d)} \leq 2y^* \|V\|_{op} (1 + y^* \|A\|_{op}) \mathcal{W}_\infty(\mu, \nu) =: \mathrm{Lip}(\mathrm{Attn}) \mathcal{W}_\infty(\mu, \nu).
$$

Finally, since

$$
\Gamma_\mu^{\mathrm{MHAttn}}(x) - \Gamma_\nu^{\mathrm{MHAttn}}(x) = W^O \begin{bmatrix} \Gamma_\mu^1(x) - \Gamma_\nu^1(x) \\ \vdots \\ \Gamma_\mu^M(x) - \Gamma_\nu^M(x) \end{bmatrix}
$$

where $\Gamma_\nu^k(x)$ denotes the mean-field self-attention of $k$-th head, we have

$$
\|\Gamma_\mu^{\mathrm{MHAttn}}(x) - \Gamma_\nu^{\mathrm{MHAttn}}(x)\|_{L^\infty(B_{t\sigma}, \mathbb{R}^d)}
$$

$$
\leq \|W^O\|_{op} \left\| \begin{bmatrix} \Gamma_\mu^1(x) - \Gamma_\nu^1(x) \\ \vdots \\ \Gamma_\mu^M(x) - \Gamma_\nu^M(x) \end{bmatrix} \right\|
$$

$$\leq \|W^O\|_{op}\sqrt{\sum_{i=1}^{M} \text{Lip}(\text{Attn}_{|\mathcal{P}(B_{t\sigma})})^2}$$

$$\leq 2y^*\sqrt{M}\|W^O\|_{op}\|V\|_{op}(1+y^*\|A\|_{op})\mathcal{W}_\infty(\mu,\nu) =: \text{Lip}(\text{MHAttn})\mathcal{W}_\infty(\mu,\nu).$$

With probability at least $\min\{1 - d/t^2, 1 - 2\exp(-s^2/(2\sigma^2))\}$, we can bound the terms by $y^* = t\sigma$, $\|W^O\|_{op}, \|V\|_{op} \leq 2\sigma\sqrt{d} + s$, $\|A\|_{op} \leq \sqrt{M/d}\|K\|_{op}\|Q\|_{op} \leq \sqrt{M/d}(2\sigma\sqrt{d} + s)^2$. Therefore, the final bounds become

$$\|\Gamma_\mu^{\text{MHAttn}}(x) - \Gamma_\nu^{\text{MHAttn}}(x)\|_{L^\infty(B_{t\sigma},\mathbb{R}^d)} \leq 2t\sigma\sqrt{M}(2\sigma\sqrt{d} + s)^2(1 + t\sigma\sqrt{\frac{M}{d}}(2\sigma\sqrt{d} + s)^2)\mathcal{W}_\infty(\mu,\nu)$$

where $M = 1$ for the single-head attention. $\qquad\square$

**Proof of Theorem 2.** We begin by bounding the Lipschitz constant for a single response $\overline{y}(\alpha)$. We denote $\overline{y}^\mu(\alpha) = \int_\mathbb{R} \left(\sum_\beta W_\beta y_i(\alpha + \beta) + b_i\right) d\mu(Wy)$, then

$$|\overline{y}^\mu(\alpha) - \overline{y}^\nu(\alpha)|$$

$$= \left|\int_\mathbb{R}\left(\sum_\beta W_\beta y(\alpha+\beta) + b_i\right) d\mu(Wy) - \int_\mathbb{R}\left(\sum_\beta W_\beta y(\alpha+\beta) + b_i\right) d\nu(Wy)\right|$$

$$= \left|\int_\mathbb{R}\left(\sum_\beta W_\beta y(\alpha+\beta) + b_i\right)(d\mu - d\nu)(Wy)\right|$$

$$\leq \left\|\left(\nabla_W\left(\sum_\beta W_\beta y(\alpha+\beta) + b_i\right), \nabla_y\left(\sum_\beta W_\beta y(\alpha+\beta) + b_i\right)\right)\right\|_2 \mathcal{W}_1(\mu,\nu)$$

$$\leq \sqrt{|\sum_\beta y(\alpha+\beta)| + |\sum_\beta W_\beta|}\, \mathcal{W}_1(\mu,\nu)$$

$$\leq \sqrt{\sum_\beta |y(\alpha+\beta)| + \sum_\beta |W_\beta|}\, \mathcal{W}_1(\mu,\nu)$$

$$\leq (2k+1)\sqrt{t\sigma + \frac{t\sigma}{(2k+1)\sqrt{C}}}\, \mathcal{W}_\infty(\mu,\nu) =: L\,\mathcal{W}_\infty(\mu,\nu).$$

Finally, since $\Gamma'_\mu(\alpha) = \overline{y}(\alpha)$, we can bound the difference between $\Gamma'_\mu$ and $\Gamma'_\nu$ as:

$$\|\Gamma'_\mu(\alpha) - \Gamma'_\nu(x)\|_{L^\infty(B_{t\sigma},\mathbb{R}^d)} = \sqrt{\sum_{i=1}^{C} |\overline{y}^\mu(\alpha) - \overline{y}^\nu(\alpha)|^2}$$

$$\leq \sqrt{C}L\,\mathcal{W}_\infty(\mu,\nu)$$

$$= (2k+1)\sqrt{t\sigma C\left(1 + \frac{1}{(2k+1)\sqrt{C}}\right)}\, \mathcal{W}_\infty(\mu,\nu).$$

$\qquad\square$

**Proof of Theorem 3.** We omit the subscript $_{|\mathcal{P}(B_{t\sigma})}$ for convenience. From Equation 2 and Lemma 1, we obtain

$$\text{Lip}^{\mathcal{W}_\infty}(\text{TF}) = \left(1 + \text{Lip}^{\mathcal{W}_\infty}(\text{MLP})\cdot\text{Lip}^{\mathcal{W}_\infty}(\text{LN})\right)\left(1 + \text{Lip}^{\mathcal{W}_\infty}(\text{MHAttn})\cdot\text{Lip}^{\mathcal{W}_\infty}(\text{LN})\right).$$

We set $x \in \mathbb{R}^d$. Since $\text{MLP}(x) = \mathbf{W}_2\phi(\mathbf{W}_1 x + \mathbf{b}_1) + \mathbf{b}_2$ and the Lipschitz constant of the ReLU activation is 1, we have

$$\text{Lip}^{\mathcal{W}_\infty}(\text{MLP}) = \|\mathbf{W}_1\| \cdot \|\mathbf{W}_2\|.$$

By Lemma 2, with probability at least $1 - 2e^{-s^2/(2\sigma^2)}$, we obtain $\|\mathbf{W}_1\|_{op}, \|\mathbf{W}_2\|_{op} \leq 2\sigma\sqrt{d} + s$. Therefore,

$$\text{Lip}^{\mathcal{W}_\infty}(\text{MLP}) \leq (2\sigma\sqrt{d} + s)^2.$$

Since $\text{LN}(x) = (x - \mu)/\sigma_x \odot \gamma + \beta$ where $\mu = \sum_{i=1}^{d} x_i/d$, $\sigma_x = \sqrt{\sum_{i=1}^{d}(x_i - \mu_i)^2/d}$, we have

$$\text{Lip}^{\mathcal{W}_\infty}(\text{LN}) = \frac{\max_i |\gamma_i|}{\sigma_x} = \frac{t\sigma}{\sigma_x}.$$

For large $d$, we approximate $\sigma_x \approx \sigma$. Specifically, by Lemma 3, with high probability, we obtain

$$|\sigma_x - \sigma| \leq \frac{\sigma\epsilon}{2}.$$

When $\sigma$ and $\epsilon$ are small, $\sigma\epsilon/2 \to 0$, leading to $\sigma_x \approx \sigma$. Therefore, using Theorem 1, the Lipschitz constant of the Transformer becomes

$$\text{Lip}^{\mathcal{W}_\infty}(\text{TF}) \leq \left(1 + t(2\sigma\sqrt{d} + s)^2\right)\left(1 + 2t^2\sigma\sqrt{M}(2\sigma\sqrt{d} + s)^2(1 + t\sigma\sqrt{\frac{M}{d}}(2\sigma\sqrt{d} + s)^2)\right),$$

where $M = 1$ for single-head attention. $\qquad\square$

**Lemma 3.** *Denote $x \sim N(0, \sigma^2 I_d)$, $\mu = \frac{1}{d}\sum_{i=1}^{d} x_i$, and*

$$y = \sqrt{\frac{1}{d}\sum_{i=1}^{d}(x_i - \mu)^2}.$$

*For any $0 < \epsilon < 1$, with probability at least $1 - 2\exp(-d\epsilon^2/8)$, we have*

$$|y - \sigma| \leq \frac{\sigma\epsilon}{2}.$$

**Proof of Lemma 3.** Since $x \sim N(0, \sigma^2 I_d)$, the sample variance satisfies

$$\sum_{i=1}^{d}(x_i - \mu)^2 \sim \sigma^2 \chi_{d-1}^2.$$

Define the chi-squared variable $Q \sim \chi_{d-1}^2$. Then, we can write

$$y^2 = \frac{1}{d}\sum_{i=1}^{d}(x_i - \mu)^2 = \frac{\sigma^2}{d}Q,$$

or equivalently, $y = \sigma\sqrt{Q/d}$.

Standard concentration results for chi-squared random variables imply that for any $0 < \epsilon < 1$,

$$P\left[\left|\frac{Q}{d-1} - 1\right| \geq \epsilon\right] \leq 2\exp\left(-\frac{(d-1)\epsilon^2}{8}\right).$$

For large $d$, replacing $d - 1$ by $d$ yields

$$P\left[\left|\frac{Q}{d} - 1\right| \geq \epsilon\right] \leq 2\exp\left(-\frac{d\epsilon^2}{8}\right).$$

Thus, with probability at least $1 - 2\exp\left(-\frac{d\epsilon^2}{8}\right)$, we have

$$\left|\frac{Q}{d} - 1\right| \leq \epsilon.$$

Now, define the function $h(u) = \sqrt{u}$ and note that its derivative is $h'(u) = \frac{1}{2\sqrt{u}}$. Since $Q/d$ concentrates around 1, we evaluate the derivative $h'(1) = \frac{1}{2}$.

Using a first-order Taylor expansion (the delta method), we obtain that

$$\left|\sqrt{\frac{Q}{d}} - 1\right| \leq \frac{\epsilon}{2}.$$

Multiplying by $\sigma$, we conclude that

$$\left|y - \sigma\right| = \sigma\left|\sqrt{\frac{Q}{d}} - 1\right| \leq \frac{\sigma\epsilon}{2}.$$

This completes the proof. $\qquad\square$

**Proof of Lemma 4.** We omit the subscript $|\mathcal{P}(B_{t\sigma})$ for convenience. From Equation 4 and Lemma 1, we obtain

$$\mathrm{Lip}^{\mathcal{W}_\infty}(\mathrm{Res}) = 1 + \mathrm{Lip}^{\mathcal{W}_\infty}(\mathrm{Conv})^3 \cdot \mathrm{Lip}^{\mathcal{W}_\infty}(\mathrm{BN})^3.$$

Using Lemma 3, with high probability, we have

$$\mathrm{Lip}^{\mathcal{W}_\infty}(\mathrm{BN}) \leq \frac{\max_i |\gamma_i|}{\sigma_B} \approx t.$$

Therefore, using Theorem 2, the Lipschitz constant of the bottleneck block becomes

$$\mathrm{Lip}^{\mathcal{W}_\infty}(\mathrm{Res}) \leq 1 + t^{9/2}(2k+1)^3 \left(\sigma C \left(1 + \frac{1}{(2k+1)\sqrt{C}}\right)\right)^{3/2}.$$

$\qquad\square$

## C   Further experimental results

### C.1   TopoLip and robustness performance

Results under impulse noise and pixelate corruption are shown in Figures 6–9. Overall, the trends observed with Gaussian noise largely hold for these corruptions. However, as the ResNet depth increases, the mCE, ECE, and accuracy drop on corrupted data decrease, which contrasts with the general trend observed in other cases.

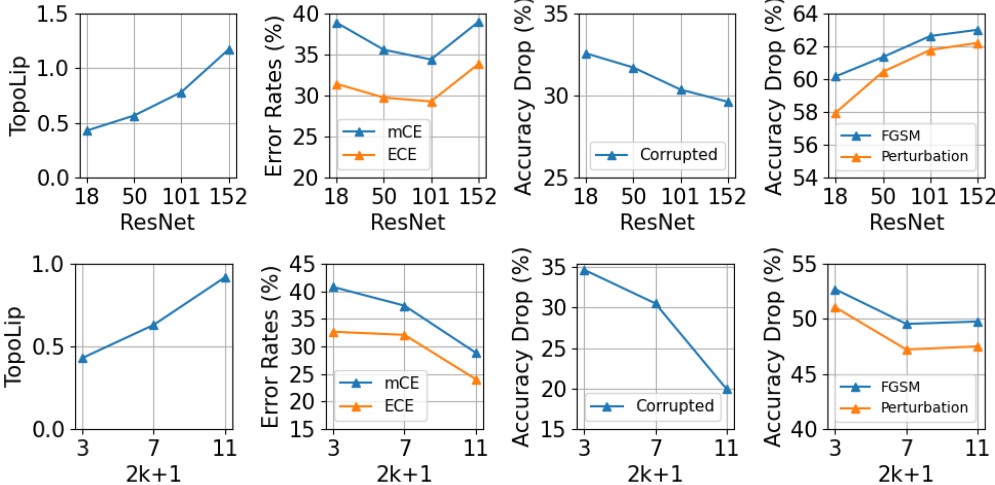

Figure 6: TopoLip and robustness performance of ResNet18 with kernel size $2k + 1 = 3/7/11$ under impulse noise corruption (CIFAR10-C). A higher ECE, mCE, or accuracy drop suggests lower robustness.

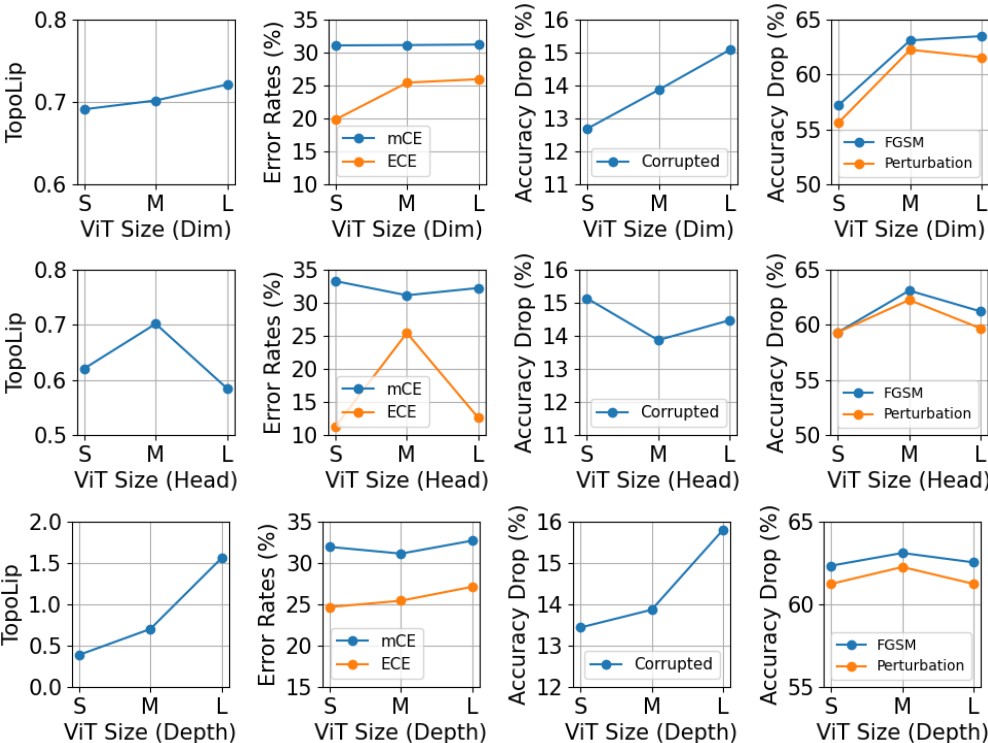

Figure 7: TopoLip and robustness performance of ViTs under impulse noise corruption (CIFAR10-C). Top row: ViTs with 12 layers and 12 heads at embedding dimensions 384 (S), 768 (M), and 1024 (L). Middle row: ViTs with 12 layers, 768 embedding dimensions, and 6 (S), 12 (M), or 16 (L) heads. Bottom row: ViTs with 12 heads, 768 embedding dimensions, and 6 (S), 12 (M), or 24 (L) layers. A higher mCE, ECE, or accuracy drop suggests lower robustness.

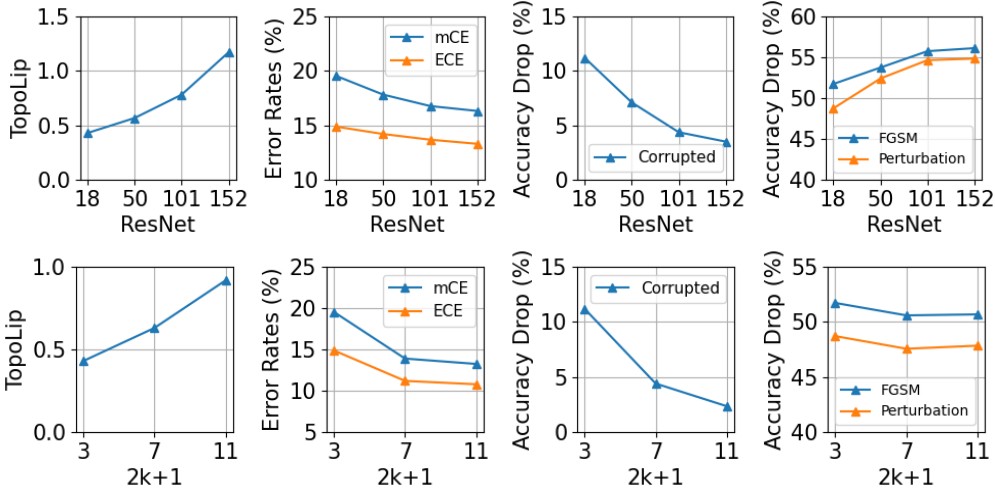

Figure 8: TopoLip and robustness performance of ResNet18 with kernel size $2k + 1 = 3/7/11$ under pixelate corruption (CIFAR10-C). A higher ECE, mCE, or accuracy drop suggests lower robustness.

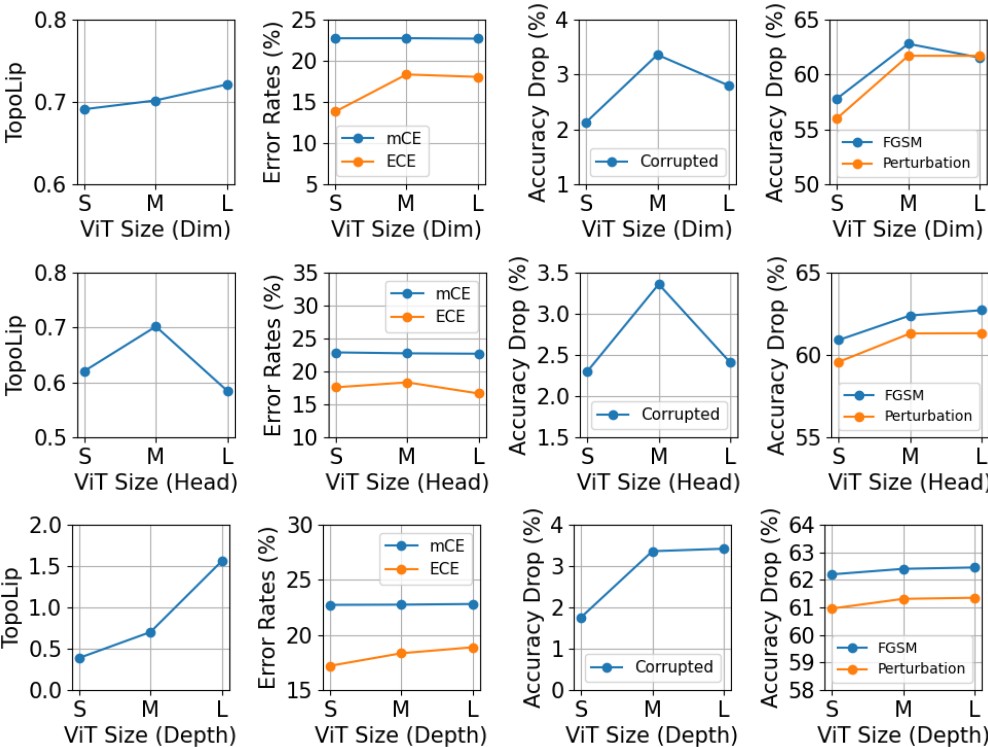

Figure 9: TopoLip and robustness performance of ViTs under pixelate corruption (CIFAR10-C). Top row: ViTs with 12 layers and 12 heads at embedding dimensions 384 (S), 768 (M), and 1024 (L). Middle row: ViTs with 12 layers, 768 embedding dimensions, and 6 (S), 12 (M), or 16 (L) heads. Bottom row: ViTs with 12 heads, 768 embedding dimensions, and 6 (S), 12 (M), or 24 (L) layers. A higher mCE, ECE, or accuracy drop suggests lower robustness.

## C.2 Training results

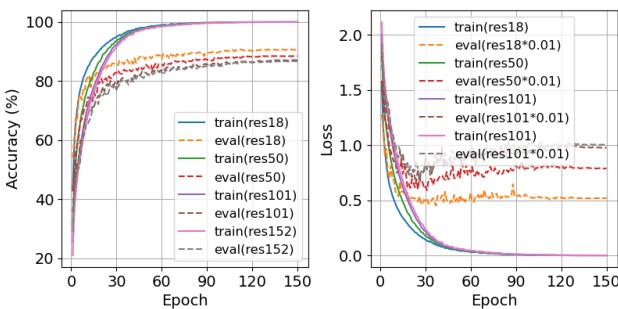

Figure 10: Accuracy and loss of ResNet18/50/101/152.

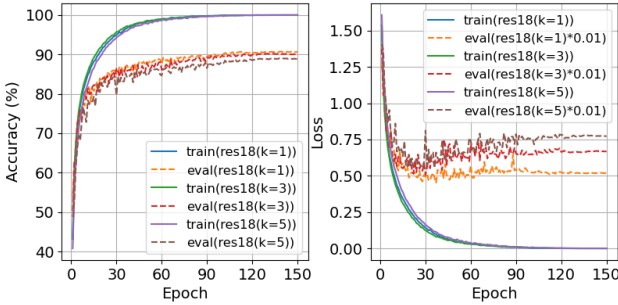

Figure 11: Accuracy and loss of ResNet18 with kernel size $2k + 1 = 3/7/11$.

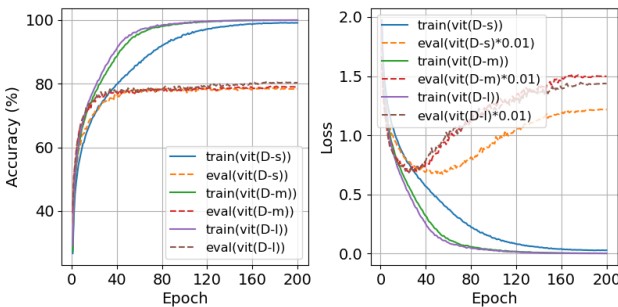

Figure 12: Accuracy and loss of ViT with 12 heads, 12 layers, and embedding dimension 384/768/1024.

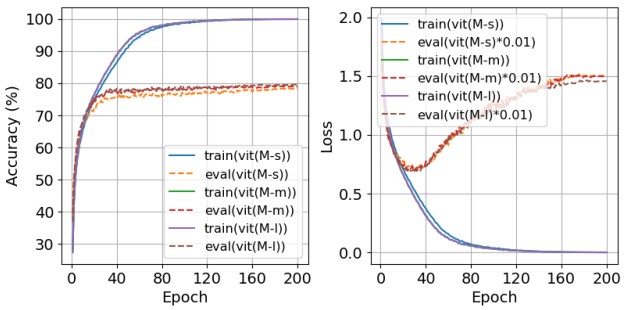

Figure 13: Accuracy and loss for a 12-layer ViT (embedding dim 768) with 6, 12, or 16 heads.

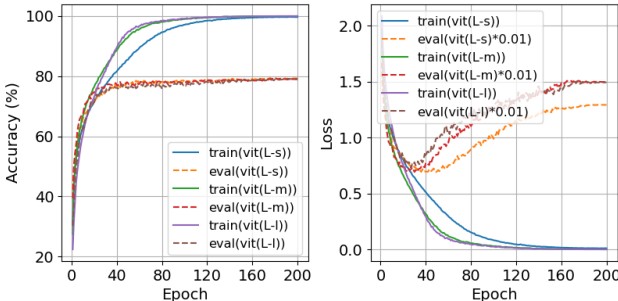

Figure 14: Accuracy and loss for a 12-head ViT (embedding dim 768) with 6, 12, or 24 layers.

