# OpenReview forum: "Is Smoothness the Key to Robustness? A Comparison of Attention and Convolution Models Using a Novel Metric"
_TMLR — Rejected by TMLR_

### Review · Reviewer_xtG5 · 2025-01-29

**Summary Of Contributions:**

This paper addresses the problem of analyzing robustness of deep neural networks. The approach uses insigths from the theory of optimal transport and Topological Data Analysis to define a novel metric for comparing the robustness of different models. The metric is based on the Lipschitz constant of the model, which is a measure of how much the output of the model changes when the input changes. The paper compares the robustness of ResNet and Vision Transformer (ViT) models on the CIFAR-10 dataset. Both experimental and theoretical results show that attention-based models are more robust than ResNet models, which is consistent with previous work. The paper also provides a theoretical analysis of the metric and shows that it can be used to compare the robustness of different models.

**Audience:**

Yes

**Broader Impact Concerns:**

Research in robustness is important, and the availability of more robust models could have a positive impact on safety, security, and reliability.

**Claims And Evidence:**

No

**Requested Changes:**

# Requests:

* The methodology assumes 'very large H x W x C' parameters, but could it be clarified how well the conclusions generalize to model sizes that actually occur in practice? (For example, the settings that are used in the experimental section.)

* The abstract mentions "Existing robustness evaluation approaches often lack theoretical generality or rely heavily on empirical assessments." However, it is not clear how this new metric relates to previous metrics? For example, can one they shown to be generalizations or specializations theoretically of one another. Or empirically, can they be shown to correlate some aspect, but provide orthogonal information in another aspect?

* Contribution "Experimental results confirm that attention-based models generally exhibit greater robustness than convolution-based models in handling corrupted data." Such conclusion was also arrived at by [1] and [2]. Can the authors clarify how their work differs from these works or does not adopt such empirical evaluation?

[1] Minderer, et al. "Revisiting the calibration of modern neural networks." Advances in Neural Information Processing Systems 34 (2021)

[2] Yung, et al. "Si-score: An image dataset for fine-grained analysis of robustness to object location, rotation and size." ICLR 2021 RobustML Workshop arXiv:2104.04191 (2021)

[3] Zaheer, et al. "Deep sets." Advances in neural information processing systems 30 (2017).


Minor nit-picks, which are not part of the review:

* Section 5: 'Lipschitness:' -> 'Lipschitzness:'

* The main text uses the terms 'permutation equivalent' and 'permutation equivariant.' It would be helpful to define these terms in the text or the appendix. It might be that this definition of 'permutation equivalent' is equivalent to what related work names 'permutation invariance,'[3] but an explicit definition would make that clear.

* Section 4.2: it is not clear where the Gromov-Wasserstein distance comes from or where it is defined. This is related literature that will be compared with?

* Section 5: quote 'a bound that is highly unpredictable' can this be clarified? What does it mean to be unpredictable in this context? I would expect that the bound is deterministic, but the actual value of the bound is unknown. Is this what is meant by unpredictable?

**Strengths And Weaknesses:**

# Strengths:

* This paper bridges the gap between optimal transport theory and topological data analysis and deep learning, which could bring new insights.

* The paper makes comparisons between ResNet architectures and ViT architectures, which are the two most-used deep neural network architectures. Any comparison between the two could help researchers help make architectural decisions.

# Weaknesses:

* It is unclear how the metric relates to a tangible outcome that matters in practise. Table 2 and Figures 2 and 3 show that the metric can be used to differentiate between models, but it is not clear how this relates to robustness in a real-world setting. For example, there is adversarial robustness, distribution-shift robustness, or robustness to perturbations such as CIFAR-C, but it is unclear how metrics in those fields related to the proposed metric. Perhaps the proposed metric introduces a new way of thinking about robustness, but then it would be helpful to have a few examples in the Appendix, especially since CIFAR is a relatively small datasets and one could plot some examples of the metric for different models.

* The argument assumes that Lipschitzness is something to be minimized, i.e., a lower value is better as indicated in Table 2. However, the reasoning is not explained and I would question it. In Image classification, for example, the difference between car and truck class could be only a threshold on the vehicle length, which is a function with infinitely high lipschitzness. Likewise, in the ImageNet dataset, the difference for some dog-breeds could be a threshold on the length of the snout, which also has high lipschitzness. Finally, in game-play, like chess or Go, one different move can exert a large difference, so the optimal value function for adjacent moves could have high lipschitzness. In all these cases, the Lipschitzness is high, but the model could still be named `robust'. So, the argument that Lipschitzness should be minimized is not clear.

---

> ### Author Response · Authors · 2025-02-20
>
> We sincerely appreciate the action editor’s detailed and constructive feedback, which has helped us refine our manuscript. Below, we address the specific concerns raised:
>
> $\textbf{Not clear how the metric relates to robustness in a real-world setting.}$
>
> In the revised version, we have extended our experiments to include other types of robustness, such as robustness to common perturbations and adversarial attacks, to provide a more comprehensive evaluation.
>
> $\textbf{Why should Lipschitzness be minimized for adversarial robustness?}$
>
> Our argument is based on the connection between Lipschitz continuity and adversarial stability. A lower Lipschitz constant means that small perturbations in the input space induce only small changes in the model’s output, thereby reducing the model’s susceptibility to adversarial examples.
>
> $\textbf{Addressing the counterexamples}$
>
> The counterexamples provided—such as classification based on thresholding vehicle length or snout length—illustrate cases where high Lipschitzness is necessary for class discrimination, but they do not necessarily imply robustness to adversarial perturbations. While certain tasks may inherently require large sensitivity to input variations (e.g., distinguishing dog breeds based on minor features), such high Lipschitzness does not contribute to robustness against adversarial noise. Instead, it often makes the model more vulnerable to adversarial attacks because even small, imperceptible perturbations can cause large changes in the model’s predictions.
>
> Additionally, in game-playing scenarios like chess or Go, high Lipschitzness in value functions may reflect the impact of critical moves, but this does not contradict our claim regarding adversarial robustness in classification tasks. In our setting, adversarial robustness refers to the model's ability to resist small adversarial perturbations in the input space, and a lower Lipschitz constant helps achieve that.
>
> $\textbf{How well does the setting “very large $H\times W\times C$” generalize to model sizes that actually occur in practice?}$
>
> We briefly discuss the appropriate parameter scaling in Section 5 (right before Section 5.1). In short, our theoretical framework applies to realistic model sizes commonly used in practical scenarios, as large $H \times W \times C$ configurations can be practically achieved with standard image datasets.
>
> $\textbf{How the work differs from [1] and [2]?}$
>
> Unlike [1] and [2], which focus on empirical comparisons, our primary contribution is the proposed metric that theoretically quantifies model robustness based on its Lipschitz properties. Our framework provides a principled way to evaluate and predict a model’s robustness before conducting extensive adversarial or corruption-based testing. While we include empirical experiments on CIFAR-10 and CIFAR-10-C to validate our metric, our goal is not to conduct large-scale empirical comparisons.
>
> $\textbf{Permutation equivalent}$
>
> We have included a brief definition of permutation equivalent before Definition 3.
>
> $\textbf{The Gromov-Wasserstein distance}$
>
> The Gromov-Wasserstein distance is a widely used method for computing the Wasserstein distance between datasets of different dimensions. In our paper, we note that it is computationally expensive and sensitive to small perturbations, which is one of the motivations for proposing TopoLip. However, a detailed comparison between Gromov-Wasserstein distance and TopoLip is beyond the scope of this work, as our focus is on defining a novel robustness metric.
>
> $\textbf{"Highly unpredictable"}$
>
> We have revised this phrase to “the bound depends on C”.
>
> We appreciate the valuable feedback and have made revisions accordingly.
>
> [1] Minderer, et al. "Revisiting the calibration of modern neural networks." Advances in Neural Information Processing Systems 34 (2021)
> [2] Yung, et al. "Si-score: An image dataset for fine-grained analysis of robustness to object location, rotation and size." ICLR 2021 RobustML Workshop arXiv:2104.04191 (2021)

---

### Review · Reviewer_hLpR · 2025-02-01

**Summary Of Contributions:**

Develops a metric for how much a the action of a model can expand the magnitude of the output. Following the literature, lower values of this quantity are more "robust". The innovation of this measure is that it can be directly compared across models of different architectures (primitive blocks) and sizes (of intermediate layers) in terms of very elementary quantities like the weight matrix magnitude, dimension, and number of heads / convolutional filters. They use this new metric to compare convolution-based and attention-based classifiers. This bound is based on mean-field approximations, and so holds with high probability for large model sizes. They find that this bound tends to be lower for attention-based architectures under typically-observed hyperparameterizations. Methodologically, the metric is based on the Wasserstein distance beween probability distributions and adopts tools from topological data analysis. Finally, they present some experiments that demonstrate the relevance of the bound by showing that training small attention and convolution-based models such as, respectively, vision transformers and resnets.

**Audience:**

Yes

**Claims And Evidence:**

Yes

**Requested Changes:**

Section 3.1:
What is W_{::}?

Section 3.3: "Lipschitzness" is not a helpful term, especially when you later sometimes use the term "Lipschitz constant" to mean the actual Lipschitz constant, and various upper bounds on the Lipschitz constant (elsewhere called "Lipschitz Bound"). Maybe different authors wrote different parts of the paper and terminology was never totally settled? The first sentence of this section should be made much clearer anyway, as it's one of the first mentions of "lipschitz" anywere in the paper.

You say "input sequence length" like it's definitely an ordered sequence, but your only actual application + the usual motivation for robustness is in images. Why not just say "input dimension"?

"Ignoring the ReLU to ease the analysis" is kind of smelly, since it only decreases the Lipschitz constant (bound). You include the nonlinearity for attention, and it seems to be important, so to not include it for conv nets seems like it could be bad. It's a bit pernicious because it's commonly done when comparing different parameterizations of the same architecture, but since you're comparing very different architectures, it's not clear that it's an innocuous choice.

Definition 5: The definition of "ker" has some problems (can sort of guess what is meant from Definition 2, but please fix).


Section 4.2:
Doesn't the Wasserstein distance really needs both distributions to live on the same space, not just the same-dimensional space?

The sentence beginning "dimension reduction techniques can align their dimensions..." is kind overly long and confusing. Would break it up and make it less redundant.

What is H_k in Definition 6?

I'm afraid that I don't exactly understand the point of defining topolip? Apparently it's proportional to the lipschitz constant. If that's it, IMO it could be in an appendix.

Section 5 It really should be "p-Wasserstein distance". Should make it clear that $\mu, \nu$ are arguments to W_p.

How does W_1 <= W_2 indicate that 1-Wasserstein Lipschitzness can be extended to 2-Wasserstein Lipschitzness? We usually want to upper-bound the Lipschitz constant?

Should it be F_{|\mathcal{X})? With the vertical line to indicated "restricted to this domain"?

So do the bounds only hold at initialization? This is a key question that deserves some attention because training is a very deliberate if not so well understood process, and almost anything could happen between initialization and convergence. More generally, it seems like the standard deviation is the overall notion of size -- nothing there directly ties to initialization -- so it might be interesting to know if the interpretation can be pushed further. It seems like the normality is just used for random matrix theory singular value bounds, so that could possible be replaced with other approaches whilst keeping a lot of the analysis intact. Might be interesting to see how far things could be generalized.

Section 5.1: It's fairly clear that ||A||_{op} is the operator norm, but best to be clear.

The second term of the product in Theorem 1 looks wrong since the M-head attention map with M = 1 does not reduce to the first equation. I'm guessing that it's down to W^O, but it just looks weird.

You could force W^0 to be the tiled identity (maybe scaled by M or square root M or whatever), without much loss of generality, then MHA with M = 1 becomes Attn.

Theorem 2: "Lipchitz" is misspelled.

"We assume H, W, C in the input image size H × W × C are very large." How large? Is 228 * 228 * 3 large?

"Furthermore, if the bound of LipW1 is tight enough, it can represent the scale or dynamics LipW1" has some problems.

I have a pretty bad intuition for how and when mean-field approximations hold. It would be good to get some further argument for why this regime applies.

Would be good to show how the theoretical bounds relate to different notions of size (# of heads, # of channels) and also noise. This should be relatively easy, just run the same experiments for different settings and plot the empirical robustness.

Section 5.2:
Where do these equations come from? The way it's written I expect to see the calculation in the appendix, but it doesn't seem to be there?
It should be Lipschitz constants (upper) _bounds_. The product of two function's lipschitz constants is an upper bound on the Lipschitz constant of the composition.

Section 6:
What does it mean that a model's "loss failed to record after the first epoch"?

Table 1 caption: Should be "TopoLip" (not "Topolip")?
Cifar 10C paper needs to be cited.

My intuition for cifar10 c is not good, but simply for cifar 10, the results do not look good. I understand that accurace per se is not the point, but it's just difficult to believe results for models that do poorly on the base problem, which itself is not hard. Ideally, you'd first establish that all models adequately solve CIFAR10, so that there's not this big question hanging over the results.

The plots would be friendlier with larger text. For Figures 3 and 4 I think you want the X axis label to be something like "cumulative depth" or something. Something went wrong with the X axis labels for Figure 1. Same with the corresponding figures in the appendix.

Section 7:
"TopoLip is effective..." without a fixed fact to point is boastful and not scientific. E.g. I expect this phrasing to describe an algorithm that provably runs faster than the SOTA or something. "TopoLip can be used for..." is true.

Appendix A:

Altogether I do not find this to be very intuitive, would probably prefer a formal intro as someone who knows nothing about the field.
The birth-death thing is not clear at all, which is kind of a problem since it features in Definition 6.

Should "D_g" be typeset differently?
"Contruction" is misspelled.


Appendix B:
Lemma 2: You want P[s_1(A) < ...]


To summarize, the paper would be much improved if:
  - Motivate the relevance of a mean-field approximation better. E.g. talk informally about the deep-learning situations in which it holds relatively well.
  - Improve the experimental component of the paper, e.g.
    - Do some adversarial attacks
    - Look at other downstream consequences of lower Lipschitz constants (e.g. overfitting, parameter stability)
    - Some sort of analysis of initialization noise.
    - Formulate a notion of the mean field approximation holding (some sort of central limit theorem) and quantify how this scales with more channels, including some infinite-width baseline.
    - Some sort of apples-to-apples comparison with standard, almost-free, methods for reducing Lipschitz constants. E.g. at least spectral normalization if not some of the more fancy methods from the recent literature based on penalizing Lipschitz constant-like regularization terms. The audience wants to know that the better robustness of attention-based models isn't easily explained or subsumed by some standard approach.
    - A fair comparison of the bound to existing approaches for upper-bounding the Lipschitz constants, such as "SeqLip" from "Lipschitz regularity of deep neural networks: analysis and efficient estimation".
  - Improve the presentation of the figures, they are simply hard to read and basically buggy.

If all or most of these are done I think it could be a great paper. Though personally, I wish the authors had done some more work to polish the paper before submission, and this should be considered some demerit.

**Strengths And Weaknesses:**

This is an interesting paper with potential, though it is still immature. Most importantly, the question asked is interesting and the approach is reasonable. However, the execution could be improved.

Overall comments:
  I find the sometimes informal and hand-wavey English incongruent with the quite formal math. I give many examples below, and would generally encourage the authors to collect feedback and critical proofreading from colleagues.
  The poor integration of the notation is problematic, e.g. the symbol W is extremely overloaded, representing at least (1) the width in a conv filter, (2) the Wasserstein distance operator, (3) a weight matrix in the definition of MLP, (3) the output projection in MHA, and (4) two different terms in Definition 5. Bolding and sub/super-scripting doesn't even fully resolve all the ambitguities.
  The relation to the other literature is also a bit rocky: for example why talk about {delta, eta}-strong robustness without having a corresponding "weak" notion? I understand that is what previous work called it, but unless you intend to dig into the nuance, it's more ergonomic to just be simpler.
  Broadly, though, the tone of the paper is good, not overly pedantic nor too lightweight. The presentation is mostly effective.

---

> ### Author Response · Authors · 2025-02-20
>
> We sincerely appreciate the action editor’s detailed and constructive feedback, which has helped us refine our manuscript. Below, we address the specific concerns raised:
>
> $\textbf{The definition of TopoLip could be in an appendix.}$
>
> TopoLip is indeed proportional to the Lipschitz constant. However, we use only a small portion of space to define it, with more emphasis on explaining the reasoning behind its definition. We believe it is important to first justify why we define it this way before formally presenting the definition.
>
> $\textbf{We usually want to upper-bound the Lipschitz constant?}$
>
> Yes, the primary purpose of TopoLip is to provide an upper bound on the Lipschitz constant.
>
> $\textbf{Do the bounds only hold at initialization?}$
>
> Our current analysis focuses on bounds at initialization, as this allows for a clean theoretical formulation using random matrix theory. However, we acknowledge that training can significantly alter model properties. While we aim to extend our results beyond initialization in future work, insights from Neural Tangent Kernel theory suggest possible directions: In the infinite-width limit, NTK theory suggests that networks retain certain properties from initialization throughout training. However, for finite-width models, the NTK evolves, which may impact robustness.
>
> $\textbf{How large should $H \times W \times C$ be?}$
>
> We briefly discuss appropriate parameter scaling in Section 5 (right before Section 5.1).
>
> $\textbf{Reasons for applying mean-field approximations.}$
>
> The reasoning behind applying mean-field approximations is included at the start of Section 3.3.
>
> $\textbf{How the theoretical bounds relate to different notions of size?}$
>
> In the revised version, we have extended our experiments to include other types of robustness, such as robustness to common perturbations and adversarial attacks, to provide a more comprehensive evaluation.
>
> $\textbf{Section 5.2: Where do these equations come from?}$
>
> We have provided proofs for these equations in the revised version.
>
> $\textbf{What does it mean that a model's ``loss failed to record after the first epoch"?}$
>
> This sentence referred to our analysis of attention-only models. In the revised version, we have omitted this analysis and instead focused on ViTs and ResNets, as attention-only and convolution-only models are rarely used in practice.
>
> $\textbf{Additional Revisions and Improvements}$
>
> In the revised version, we also made the following refinements:
>
> $\bullet$ Standardized notation for $W$ to improve clarity:\
>     1. Width in a convolutional filter: $W$.
>     2. Wasserstein distance operator: $\mathcal{W}$.
>     3. Weight matrix in MLP definition: $\mathbf{W}$.
>     4. Output projection in MHA: $W^O$.
>     5. Terms in Definition 5: $\mathtt{W}$.
>
> $\bullet$ Included a description of $W_{::}$
>
> $\bullet$ Revised "Lipschitzness" to "Lipschitz constant" where appropriate
>
> $\bullet$ Redefined "ker" for better clarity
>
> $\bullet$ Added ReLU analysis for a more comprehensive discussion
>
> $\bullet$ Refined the description of the Wasserstein distance
>
> $\bullet$ Revised a key sentence on dimension reduction techniques for better readability
>
> $\bullet$ Included the definition of $H_k$
>
> $\bullet$ Clarified that $\mu, \nu$ are arguments to $W_p$
>
> $\bullet$ Revised notation: changed $F_{|\mathcal{X}}$ for clarity
>
> $\bullet$ Explicitly stated that $||A||_{\text{op}}$ represents the operator norm
>
> $\bullet$ Aligned results for single-/multi-head attention in Theorem 1
>
> $\bullet$ Cited the CIFAR-10-C paper
>
> $\bullet$ Provided a more formal and comprehensive introduction to TDA in Appendix A
>
> $\bullet$ Improved figure presentation for better readability
>
> We appreciate the valuable feedback and have made revisions accordingly.

---

### Review · Reviewer_cvDj · 2025-02-05

**Summary Of Contributions:**

Adversarial robustness is one of the critical directions in modern machine learning.
However, most evaluations of adversarial robustness still rely on empirical experiments.
Therefore, the author proposed a theoretical framework to analysis the adversarial robustness of the model based on its architectural, and uses his framework proved that the attention-based models are more robust than convolution-based models.

**Audience:**

Yes

**Claims And Evidence:**

Yes

**Requested Changes:**

I personally suggest that the term ``robustness'' is too broad, as the author is specifically focusing on adversarial robustness.
It would be clearer to explicitly use the term 'adversarial robustness' instead.

**Strengths And Weaknesses:**

Strengths:
+This work is interesting and provides a useful theoretical framework to measure the adversarial robustness based on the architecture of the model.

Weaknesses:
-The methodology requires a strong understanding of topological concepts (e.g., persistence diagrams), which could limit its practical applicability.

-The framework can only measure adversarial robustness based on the layers used by the model, but it cannot capture differences between models that share the same architecture. Notably, while model architectures are limited, many research efforts focus on improving robustness within existing architectures. However, this framework cannot reflect robustness improvements made to models with the same architecture

---

> ### Author Response · Authors · 2025-02-20
>
> We sincerely appreciate the constructive feedback, which has helped us refine our manuscript. Below, we address the specific concerns raised:
>
> 1. Capturing Differences Between Models with the Same Architecture We agree that robustness can vary even among models with the same architecture due to differences in training strategies and parameter choices. However, our framework is designed to account for such variations. Theoretically, we show that the Lipschitz constant depends on architectural hyperparameters such as depth ($L$), number of attention heads ($M$), and embedding dimension ($d$). These parameters directly influence the robustness of different configurations within the same model family. Experimentally, we validate this by comparing Vision Transformers (ViTs) with varying $L$, $M$, and $d$, demonstrating that our framework successfully distinguishes robustness differences across these variations.
> 2. Clarifying the Use of "Robustness" We acknowledge the concern that "robustness" is a broad term. In the revised version, we have extended our experiments to include other types of robustness, such as robustness to common perturbations and adversarial attacks, to provide a more comprehensive evaluation.
>
> We appreciate the valuable feedback and have made revisions accordingly.

---

### Decision · Action_Editor_Yq1D · 2025-03-17

**Recommendation:** Reject

**Comment:**

While the topic holds interest for the TMLR audience, the evidence presented is insufficient to support the claims. Therefore, I recommend rejecting this paper in its current form.

However, I recommend that the authors work on improving their manuscript and resubmit a Major Revision at a later time. I recommend that the authors significantly expand the experimental evaluation to demonstrate a clear causal link between TopoLip and robustness, thoroughly comparing it with existing robustness metrics across diverse settings and highlighting what new insights it unlocks. Furthermore, I suggest they provide a more rigorous and clearly defined scope for the notion of robustness addressed, potentially narrowing the focus for clarity.

**Audience:**

All reviewers agree that the topic of this paper - expanding the set of evaluation metrics and theoretical insights about the robustness of neural networks - is an area of interest to the TMLR audience. In particular, the use of notions from Topological Data Analysis and Optimal Transport to compute Lipschitz constants of neural networks presents an interesting research avenue that could be of interest to some researchers within the community.

**Claims And Evidence:**

The central claim of this paper is that TopoLip offers a more general and principled approach to assessing the robustness of neural networks compared to existing methods. While the reviewers and I appreciate the motivation behind this work, we find that the evidence presented in the submission is not enough to support its claims.

Specifically, a majority of reviewers agree that the experimental evaluation is too limited in scope and does not compare favorably with the extensive evaluations in prior work. The analysis primarily demonstrates only partial correlations between TopoLip and some common robustness metrics and while these correlations might be interesting, they fall short of establishing a clear and direct causal relationship between TopoLip and robustness. Furthermore, the definition of "robustness" used throughout the paper is vague and conflates distinct concepts from the literature without rigorous motivation or analysis. Therefore, having read the paper myself, I agree with the reviewers that this version of the paper is not ready for acceptance at this time.

**Resubmission Of Major Revision:**

The authors may consider submitting a major revision at a later time.